# Fed-CO$_2$: Cooperation of Online and Offline Models for Severe Data Heterogeneity in Federated Learning

**Zhongyi Cai**
ShanghaiTech University
caizhy@shanghaitech.edu.cn

**Ye Shi** *
ShanghaiTech University
shiye@shanghaitech.edu.cn

**Wei Huang**
RIKEN Center for Advanced Intelligence Project
wei.huang.vr@riken.jp

**Jingya Wang**
ShanghaiTech University
wangjingya@shanghaitech.edu.cn

## Abstract

Federated Learning (FL) has emerged as a promising distributed learning paradigm that enables multiple clients to learn a global model collaboratively without sharing their private data. However, the effectiveness of FL is highly dependent on the quality of the data that is being used for training. In particular, data heterogeneity issues, such as label distribution skew and feature skew, can significantly impact the performance of FL. Previous studies in FL have primarily focused on addressing label distribution skew data heterogeneity, while only a few recent works have made initial progress in tackling feature skew issues. Notably, these two forms of data heterogeneity have been studied separately and have not been well explored within a unified FL framework. To address this gap, we propose Fed-CO$_2$, a universal FL framework that handles both label distribution skew and feature skew within a **C**ooperation mechanism between the **O**nline and **O**ffline models. Specifically, the online model learns general knowledge that is shared among all clients, while the offline model is trained locally to learn the specialized knowledge of each individual client. To further enhance model cooperation in the presence of feature shifts, we design an intra-client knowledge transfer mechanism that reinforces mutual learning between the online and offline models, and an inter-client knowledge transfer mechanism to increase the models' domain generalization ability. Extensive experiments show that our Fed-CO$_2$ outperforms a wide range of existing personalized federated learning algorithms in terms of handling label distribution skew and feature skew, both individually and collectively. The empirical results are supported by our convergence analyses in a simplified setting.

## 1 Introduction

Federated Learning (FL) [1, 2] is a distributed learning framework that involves collaboratively training a global model with multiple clients while ensuring privacy protection. The pioneering work FedAvg [2] learns the global model by aggregating the local client models and obtains satisfactory performance when the client data are independently and identically distributed (IID). However, when the data are heterogeneous and non-IID among clients, performance with the global model can degrade substantially [3, 4]. Common data heterogeneity issues include label distribution skew and feature skew. Clients experiencing label distribution skew exhibit varying class distributions within the same domain, while clients encountering feature skew maintain consistent class distributions

---

*Corresponding author

but belong to different domains. As a promising solution to data heterogeneity issues, Personalized Federated Learning (PFL) has emerged, where personalized models are trained for individual clients. Most previous studies have focused on addressing label distribution skew [5, 6, 7], while only a few recent works [8, 9] have made initial progress in tackling feature shift issues. So far, these two forms of data heterogeneity have been studied separately and have not been addressed within a unified FL framework. Therefore, in this paper, we propose a universal FL framework that effectively handles data heterogeneity issues arising from label distribution skew, feature skew, or their combination.

Several algorithms personalized some parts of the model in FL with label distribution imbalance or feature shifts to retain and leverage some of the local offline information [10, 11, 12, 13, 8]. However, in realistic FL scenarios, where extreme label distribution skew, severe feature skew, or even both are present, these algorithms fail to effectively harness local specialized knowledge for satisfactory adaptation. In fact, in some cases of extreme heterogeneity, models trained by those personalized algorithms may even perform worse than the locally-trained model, as the locally-trained model excels in capturing offline specialized knowledge. On the other hand, in FL scenarios with milder heterogeneity, partially personalized models trained by PFL algorithms perform better due to their ability to access online general information from other clients.

So, given that the model with partially personalized parameters and the locally trained model each perform better in different cases, the question is: Is there a more effective approach to fuse the online general knowledge and the offline specialized knowledge for better performance? From our investigations, the answer is yes. Accordingly, we propose a novel universal cooperation framework with these two models to address this challenge for both label distribution skew and feature skew data heterogeneity, referring to the model with partially personalized parameters as the online model, and the locally trained model as the offline model. Specifically, we personalize Batch Normalization layers in the online model and fuse the online and offline models' predictions as the final prediction. The prediction fusion between the online and offline models rectifies errors in their respective individual predictions and exhibits better performance.

In FL scenarios with feature skew, the general knowledge learned by the online model is domain-invariant, whereas the specialized knowledge learned by the offline model is domain-specific. Cooperation that occurs solely by fusing predictions is not sufficient as this process does not let the online and offline models communicate during the training phase. In turn, this impedes the transfer of domain-invariant and domain-specific knowledge between the models. Hence, to further encourage cooperation between the models, we propose two novel knowledge transfer mechanisms - one intra-client and one inter-client - which work at the model and client level, respectively. The intra-client knowledge transfer mechanism facilitates mutual learning between the online and offline models via knowledge distillation, which enables the online and offline models to benefit from both online domain-invariant and offline domain-specific knowledge. Conversely, the inter-client knowledge transfer mechanism enhances the model's domain generalization ability by introducing classifiers from the offline models of other clients to each local client.

**Contribution.** We propose Fed-$CO_2$, a universal cooperative FL framework for severe data heterogeneity including both label distribution skew and feature skew. By simply fusing the online and offline models, Fed-$CO_2$ can handle severe label distribution skew effectively. To enhance model performance in the presence of severe feature skew, Fed-$CO_2$ involves an intra-client knowledge transfer mechanism that improves model cooperation and an inter-client knowledge transfer mechanism that increases client cooperation. We theoretically show that Fed-$CO_2$ has a faster convergence rate than FedBN [10] in a simplified setting. Besides, extensive experiments on five benchmark datasets demonstrate that our Fed-$CO_2$ framework has a prominent edge over a range of state-of-the-art algorithms where the data contain label distribution skew, feature skew, or both[1].

## 2 Related Work

**Federated Learning for Label Distribution Skew Data Heterogeneity.** In real-world FL scenarios, label distribution imbalance among clients is a common phenomenon that poses challenges to learning a single model that can effectively cater to all clients. A straightforward approach to personalizing the global model is fine-tuning it on local datasets [14, 15, 16, 17]. Other approaches attempt to overcome distribution heterogeneity by exerting a proximal regularization term on the global model,

---

[1]Our codes are publicly available at https://github.com/zhyczy/Fed-CO2

as seen in examples like FedProx [4], pFedMe [18], MOON [7], and Ditto [19]. Instead of adapting the single global model, learning personalized models is another main track. Here, the more explicit methods [11, 20, 21, 22] personalize some of the model parameters while leaving other parameters for aggregation. Similarly, FedMask [23] learns distinct binary masks for the last several layers of the local models. Further, inspired by Hypernetworks [24], some methods [6, 13, 25] apply a hypernetwork to produce client-specific parameters for participants. Beyond directly personalizing parts of the model parameters, some of the newer methods involve a two-branch architecture where the aim is to calibrate the model's predictions [12, 26, 27]. For example, FedRoD [12] uses a personal head to strengthen local learning, while PerFCL [26] splits the features in the local clients into shared parts and personalized parts. FedProc [27] uses prototypes as global knowledge to correct local training. However, unlike these methods, Fed-CO$_2$ maintains two separate personalized models for each client – one that learns online general knowledge and the other that learns offline specialized knowledge – with a focus on facilitating cooperation between the two models.

Additionally, knowledge distillation, which transfers dark knowledge between models, has played an important role in addressing label distribution skew [28, 29, 30, 31, 32, 22]. Most of previous works in this arena rely heavily on a public dataset to transfer knowledge between the server and the local clients [29, 30, 31]. Only a limited number of studies have delved into the synergistic interplay between learning global knowledge and learning local knowledge, as well as strategies to optimize the mutual benefits of these two processes. In FML [28], the global model and the local model engage in mutual learning, but only the local model is used for personalized predictions. A very recent study, CD$^2$-pFed [22], employs cyclical distillation as a regularization term in conjunction with the local training cross-entropy loss to mitigate the gaps between the learned representations from local weights and those from global weights. By contrast, our method utilizes knowledge distillation to facilitate the exchange of beneficial knowledge between the online and offline models before local training, as well as to ensure effective and adequate intra-client knowledge transfer.

**Federated Learning for Feature Skew Data Heterogeneity.** While various methods have shown promising results on label distribution data heterogeneity, they often suffer from significant performance degradation when confronted with the more challenging feature shift data heterogeneity. To adapt to the feature shifts among different domains, FedBN [10] personalizes Batch Normalization (BN) layers for each participant. Beyond personalizing BN layers, PartialFed [8] explores the strategy of selecting personalized parameters based on the feature characteristics of different clients. To handle the feature skew, FedAP [9] learns the similarities between participants by analyzing their private parameters in the BN layers. In a recent work called TrFedDis [33], the concept of feature disentangling is used to capture both domain-invariant and domain-specific knowledge. Our Fed-CO$_2$ deviates from such disentangling approaches and instead focuses on facilitating mutual learning between the online and offline models.

# 3 METHOD

In this section, we begin by formulating the research problem and subsequently introduce Fed-CO$_2$, our novel universal FL framework. Finally, we will present the intra-client and inter-client knowledge transfer mechanisms designed to further enhance model cooperation in FL with feature skew.

## 3.1 Problem Formulation

We aim to train a set of $N$ models for each client to fit local data distribution. Each client $i$ has its own data distribution $P_i$ with its private dataset $D_i = \{(x_i^k, y_i^k) : k \in \{1, \ldots, m_i\}\}$, $i \in \{1, \ldots, N\}$ and $m_i$ is the sample number in the private dataset. Due to label distribution skew or feature skew, the data distribution $P_i(x, y)$ differs on a client-by-client basis. Our goal is to learn a good model $F_i(\cdot)$ with parameters $\theta_i$ for each client according to information in all clients:

$$\min_{\{\theta_i\}_{i=1}^N} \frac{1}{N} \sum_{i=1}^N \sum_{j=1}^{m_i} l(F_i(\theta_i; x_i^j), y_i^j), \tag{1}$$

where $l(\cdot, \cdot)$ is a sample-wise loss function, $F_i$ is constructed by a feature extractor $f_i(\cdot)$ with parameters $\eta_i$ and a classifier $C_i(\cdot)$ with parameters $\phi_i$. Accordingly, $\theta_i = \{\eta_i, \phi_i\}$.

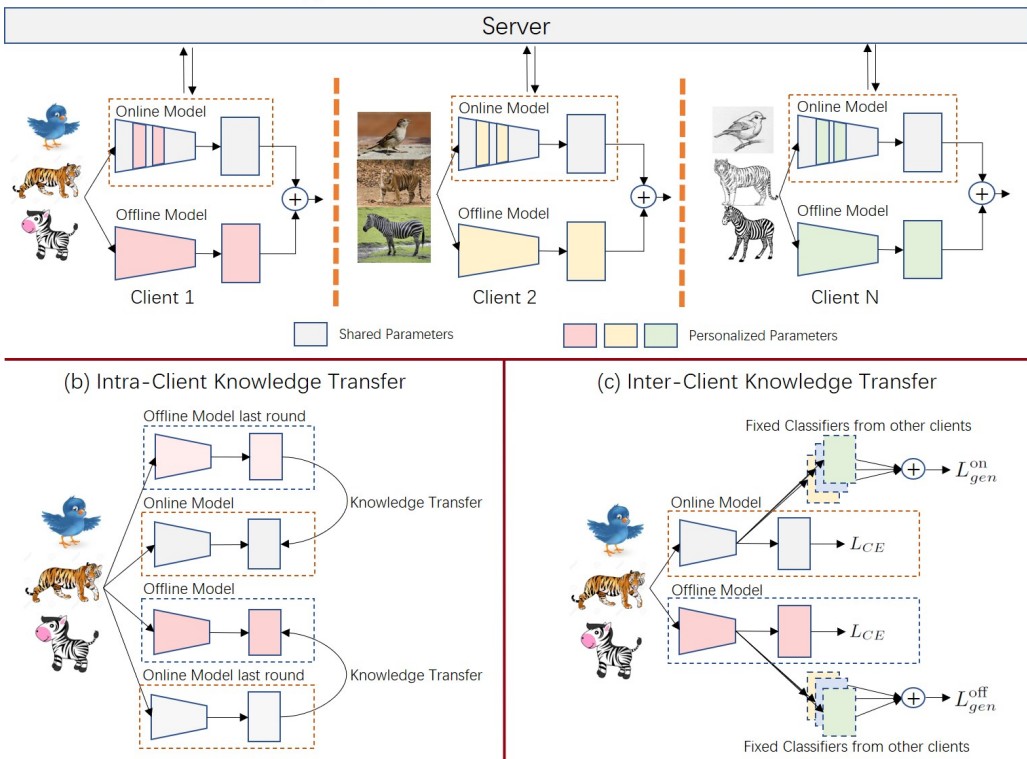

Figure 1: The Fed-CO$_2$ framework. Fig. 1(a) shows an overview of the cooperation between the online and offline models in Fed-CO$_2$. Figures 1(b) and 1(c) respectively illustrate the intra-client and inter-client knowledge transfer mechanisms on FL with feature skew.

## 3.2 Cooperation of Online and Offline Models

As shown in Fig. 1(a), we propose Fed-CO$_2$: a novel universal collaboration framework between the online and offline models to adapt to local data distribution in the presence of label distribution skew, feature skew, or both. Specifically, for each client $i$, we train a partially personalized model, referred to as the online model $F_i^{\text{on}}$, and a locally-trained model, referred to as the offline model $F_i^{\text{off}}$. With two models in each client $F_i = \{F_i^{\text{on}}, F_i^{\text{off}}\}$, cooperation is achieved through prediction fusion of the online and offline models. In this manner, our framework aims to exploit both online general and offline specialized knowledge, enabling consistent high performance in FL across various cases of local data heterogeneity.

For the online model, we personalize only a few critical parameters to enable the online model to learn knowledge from other clients through model aggregation on the server, while still retaining essential local knowledge. Prior work FedBN [10] has shown that Batch Normalization (BN) layers can capture feature distribution in local clients. Inspired by such discovery, we personalize the BN layers in the online model and split its parameters $\theta_i^{\text{on}}$ into $\{\theta_{p,i}^{\text{on}}, \theta_g^{\text{on}}\}$, where $\theta_{p,i}^{\text{on}}$ denotes all BN layer parameters and $\theta_g^{\text{on}}$ denotes other parameters. Model aggregation is done on the server to obtain the shared $\widetilde{\theta}_g^{\text{on}}$ among all clients with the formula:

$$\widetilde{\theta}_g^{\text{on}} = \sum_{i=1}^{N} \frac{1}{N} \theta_{g,i}^{\text{on}}. \tag{2}$$

For the offline model, all its parameters are personalized and do not engage in server aggregation. As a result, the offline model is able to learn local specialized knowledge without forgetting or being contaminated by other irrelevant information. With online and offline models, we employ a simple fusion technique and obtain the final prediction for model $F_i$:

$$F_i(\theta_i \, ; \, x) = F_i^{\text{on}}(\theta_{p,i}^{\text{on}}, \, \theta_g^{\text{on}} \, ; \, x) + F_i^{\text{off}}(\theta_i^{\text{off}} \, ; \, x). \tag{3}$$

It's worth noting that the online-offline model cooperation does not require extra communication costs compared to standard federated aggregation operations since offline models are not uploaded to the server.

When feature skew data heterogeneity is present, relying solely on prediction fusion for cooperation is insufficient. This approach fails the communication between the online and offline models during the training phase, impeding the transfer of online domain-invariant and offline domain-specific knowledge. Therefore, to enhance collaboration among the models in FL with feature skew, we propose novel intra-client and inter-client knowledge transfer mechanisms. In the next two parts, we will introduce these mechanisms and discuss their role.

### 3.3 Intra-Client Knowledge Transfer

In our framework, each client $i$ is equipped with two models: an online model for capturing general domain-invariant knowledge and an offline model for capturing specialized domain-specific knowledge. Unlike previous algorithms that utilized a two-branch framework, aiming to make the knowledge learned by each branch as disjoint as possible through methods such as Contrastive Learning [26] or Feature Disentangling [33], we introduce a novel intra-client knowledge transfer mechanism to enhance the collaboration of the two models. This mechanism employs knowledge distillation to facilitate mutual learning between the two models during the training phase. However, transferring online general knowledge to the offline model could potentially conflict with the local training objective. The former emphasizes learning global information among $\{D_j\}^{j \neq i}$ and avoiding over-fitting on local data, whereas the latter focuses on fitting local information in $D_i$. As a result, neither of the two goals can be fully accomplished. Furthermore, during local training, there is a risk of the online model losing general domain-invariant knowledge, which leads to a reduction in the amount of valuable information accessible to the offline model.

To address these challenges, we divide the local training process into two distinct phases: a mutual learning phase and a local adaptation phase to achieve their respective goals. Concretely, in the mutual learning phase, we create a duplicate of the initial online model, which consists of its personalized BNs from the last round and other parameters updated with the global aggregation model. Additionally, we retain a copy of the initial offline model from the last round. As illustrated in Fig. 1(b), these duplicated models are frozen and utilized as teacher networks, with the copied online model serving as a teacher for the offline model, and the copied offline model guiding the online model. We denote the duplicated initial online model and copied initial offline model as $\bar{F}_i^{\text{on}}$ and $\bar{F}_i^{\text{off}}$, respectively, with frozen parameters $\bar{\theta}_i^{\text{on}}$ and $\bar{\theta}_i^{\text{off}}$. In training step $s$, given input sample $x_k$, online and offline models conduct intra-client knowledge transfer and update their parameters in the mutual learning form:

$$\theta_{i,s}^{\text{on}} = \theta_{i,s-1}^{\text{on}} - \alpha \cdot \nabla_{\theta_{i,s-1}^{\text{on}}} KL(\bar{F}_i^{\text{off}}(\bar{\theta}_i^{\text{off}}\,;\,x_k)\,,\,F_i^{\text{on}}(\theta_{i,s-1}^{\text{on}}\,;\,x_k)), \tag{4}$$

$$\theta_{i,s}^{\text{off}} = \theta_{i,s-1}^{\text{off}} - \alpha \cdot \nabla_{\theta_{i,s-1}^{\text{off}}} KL(\bar{F}_i^{\text{on}}(\bar{\theta}_i^{\text{on}}\,;\,x_k)\,,\,F_i^{\text{off}}(\theta_{i,s-1}^{\text{off}}\,;\,x_k)), \tag{5}$$

where $KL(\cdot, \cdot)$ denotes the Kullback-Leibler (KL) divergence and $\alpha$ is the learning rate. Through mutual learning, the online model and the offline model engage in model-level collaboration and share the knowledge acquired in the previous round.

### 3.4 Inter-Client Knowledge Transfer

After the intra-client knowledge transfer in the first phase, both the online and offline models have gained beneficial knowledge from each other, and now it is opportune to let them adapt to the local data distributions. The pertinent question is whether we can leverage additional knowledge from other clients to bridge the domain gaps in the local training. Currently, the approach of acquiring knowledge from other clients is limited to aggregating shared parameters on the server. In comparison to various advanced approaches used in Domain Generalization (DG) research, parameter aggregation alone may struggle to learn effective domain-invariant features. The primary challenge stems from the isolation of private data, as FL prohibits cross-client access to data from other domains. Without data from multiple domains, it is tough to directly apply techniques in DG.

Considering that the offline model in each client is trained to capture local specialized domain-specific knowledge, we propose to leverage these lightweight classifiers of the offline model to transfer domain

knowledge among clients, as illustrated in Fig. 1(c). The classifiers of the offline model from other clients are introduced to each local client, enabling the feature extractors of its online and offline models to generate robust and well-generalized features that can be effectively recognized by these introduced classifiers. We share similar design intuitions with COPA [34], but here our inter-client knowledge transfer mechanism stresses more on making use of knowledge from other clients to benefit the personalized model for each client rather than serving unseen novel clients.

To be specific, in Fed-CO$_2$, each client uploads its classifier $C_i^{\text{off}}$ from its offline model to the server, along with its shared parameters (BNs are not included) from its online model. These uploaded heads construct a classifier set for inter-client knowledge transfer: $\{\overline{C}_j^{\text{off}}\}_{j=1}^N$ with parameter set $\{\overline{\phi}_j^{\text{off}}\}_{j=1}^N$ and are transmitted to each client in the next round. In communication round $t$, each client has its own personalized model $F_i = \{F_i^{\text{on}}, F_i^{\text{off}}\}$ and the downloaded knowledge-transfer classifier set $\{\overline{C}_j^{\text{off}}\}_{j=1}^N$. Similar to teacher models in the mutual learning phase, we freeze these knowledge-transfer classifiers to keep their knowledge unchanged. With input data $x_k$ and its label $y_k$ our inter-client knowledge transfer loss function $L_{gen}$ for models on client $i$ is formulated as:

$$L_{gen}\left(x_k\,,\,y_k\right) = \sum_{j \neq i} L_{CE}\left(\overline{C}_j^{\text{off}}\left(\overline{\phi}_j^{\text{off}}; f_i\left(\eta_{i,t}; x_k\right)\right),\, y_k\right), \tag{6}$$

Combined with its classification loss to adapt to local private data, the total loss in local adaptation phase for each model is:

$$L = L_{CE}(x_k\,,\,y_k) + \mu \cdot L_{gen}(x_k\,,\,y_k), \tag{7}$$

where $\mu$ is a penalized factor set as 1 by default. After local training, we upload parameters to the server and aggregate them with Eq. 2. Algorithm 1 in Appendix demonstrates the procedures of our Fed-CO$_2$ algorithm.

## 4 Theoretical Analysis

Here, we provide convergence analyses, which compare our Fed-CO$_2$ and FedBN [10] with the neural tangent kernel (NTK) [35] theory, to illustrate the effectiveness of our algorithm. For simplification, only the cooperation of online and offline models is considered while the intra-client and inter-client knowledge transfer mechanisms are ignored.

### 4.1 Formulation

Before our analyses, we clarify the simplified setup and basic assumptions. Suppose we have $N$ clients, each with $M$ training examples, and we aim to jointly train them for $T$ communication rounds. In each round, each model is locally trained for one epoch. For simplification, we assume that all clients share an identical two-layer neural network for a regression task, with the only source of heterogeneity being the feature skew. Namely, each client $i$ has its private dataset $\mathbb{D}_i = \{(\mathbf{x}_j^i, y_j^i)\}_{j=1}^M$ with $\mathbf{x}_j^i \in \mathbb{R}^d$ and $y_j^i \in \mathbb{R}$. For convenience, we adopt the same data distribution assumption as in [10]:

**Assumption 4.1** *For each client $i \in \{1, \ldots, N\}$, the inputs $\mathbf{x}_j^i$ are centered, meaning that $\mathbb{E}[\mathbf{x}^i] = 0$, and they have a covariance matrix $\mathbf{S}_i = \mathbb{E}[\mathbf{x}^i(\mathbf{x}^i)^\top]$. $\mathbf{S}_i$ is independent of the label $y$ and varies for each client $i$. Not all $\mathbf{S}_i$ matrices are identity matrices. For any index pair $p$ and $q$ ($p \neq q$), we have $\mathbf{x}_p \neq w \cdot \mathbf{x}_q$ for any non-zero $w$.*

Let $\mathbf{v}_k \in \mathbb{R}^d$ represent the parameters of the first layer, where $k \in [m]$ and $m$ is the width of the hidden layer. We define $\|\mathbf{v}\|_{\mathbf{S}} := \sqrt{\mathbf{v}^\top \mathbf{S} \mathbf{v}}$ as the induced vector norm for a positive definite matrix $\mathbf{S}$ and a $l_2$ vector norm $\|\mathbf{v}\|_2$. The projections of $\mathbf{x}$ onto $\mathbf{v}$ and $\mathbf{v}^\perp$ are defined as $\mathbf{x}^{\mathbf{v}} := \frac{\mathbf{v}\mathbf{v}^\top \mathbf{x}}{\|\mathbf{v}\|_2^2}$, $\mathbf{x}^{\mathbf{v}^\perp} := \left(\mathbf{I} - \frac{\mathbf{v}\mathbf{v}^\top}{\|\mathbf{v}\|_2^2}\right)\mathbf{x}$.

Based on Assumption 4.1, for client $i$, the output of the first layer is normalized as $\frac{\mathbf{v}_k^\top \mathbf{x}^i}{\|\mathbf{v}_k\|_{\mathbf{S}_i}}$. The shift parameter of BN is omitted. Further, we denote the scaling parameter of BN $\gamma \in \mathbb{R}^{m \times N}$ and the second layer parameter $c \in \mathbb{R}^m$.

With these parameters, we proceed to train the online model:

$$\mathbf{F}^{\mathrm{on}}(\mathbf{x}\,;\,\mathbf{V},\,\gamma,\,c) = \frac{1}{\sqrt{m}} \sum_{k=1}^{m} c_k^{\mathrm{on}} \sum_{i=1}^{N} \sigma\left(\gamma_{k,i}^{\mathrm{on}} \cdot \frac{\mathbf{v}_k^{\mathrm{on}^\top} \mathbf{x}}{\|\mathbf{v}_k^{\mathrm{on}}\|_{\mathbf{s}_i}}\right) \cdot \mathbb{1}\{\mathbf{x} \in \mathbb{D}_i\}, \tag{8}$$

where $\sigma(\cdot)$ is the ReLU activation function and $\gamma_{k,i}^{\mathrm{on}}$ is personalized BN parameters. For the offline model with all parameters personalized, we train:

$$\mathbf{F}^{\mathrm{off}}(\mathbf{x}\,;\,\mathbf{V},\,\gamma,\,c) = \frac{1}{\sqrt{m}} \sum_{k=1}^{m} \sum_{i=1}^{N} c_{k,i}^{\mathrm{off}} \cdot \sigma\left(\gamma_{k,i}^{\mathrm{off}} \cdot \frac{\mathbf{v}_{k,i}^{\mathrm{off}^\top} \mathbf{x}}{\|\mathbf{v}_{k,i}^{\mathrm{off}}\|_{\mathbf{s}_i}}\right) \cdot \mathbb{1}\{\mathbf{x} \in \mathbb{D}_i\}. \tag{9}$$

With online and offline models, we form our Fed-CO$_2$ as model $\mathbf{F}$:

$$\mathbf{F}(\mathbf{x}\,;\,\mathbf{V},\,\gamma,\,c) = \frac{1}{2}\left(\mathbf{F}^{\mathrm{on}}(\mathbf{x}\,;\,\mathbf{V}^{\mathrm{on}},\,\gamma^{\mathrm{on}},\,c^{\mathrm{on}}) + \mathbf{F}^{\mathrm{off}}(\mathbf{x}\,;\,\mathbf{V}^{\mathrm{off}},\,\gamma^{\mathrm{off}},\,c^{\mathrm{off}})\right). \tag{10}$$

In our analysis, we adopt one random strategy [36] to initialize parameters:

$$\mathbf{v}_{k,i}^{\mathrm{off}}(0) = \mathbf{v}_k^{\mathrm{on}}(0) \sim \mathcal{N}(0,\,\alpha^2\mathbf{I}); \; c_{k,i}^{\mathrm{off}} = c_k^{\mathrm{on}} \sim \mathrm{Unif}\{-1,\,1\}; \; \gamma_{k,i}^{\mathrm{on}} = \gamma_{k,i}^{\mathrm{off}} = \frac{\|\mathbf{v}_k^{\mathrm{on}}(0)\|_2}{\alpha}, \tag{11}$$

where $\alpha^2$ controls the magnitude of $\mathbf{v}_k^{\mathrm{on}}$ and $\mathbf{v}_{k,i}^{\mathrm{off}}$ at initialization. The initialization of the BN parameters $\gamma_{k,i}^{\mathrm{on}}$ and $\gamma_{k,i}^{\mathrm{off}}$ are independent of $\alpha$. We use the MSE loss to train our model $\mathbf{F}$:

$$\begin{aligned} L(\mathbf{F}) &= \frac{1}{NM} \sum_{i=1}^{N} \sum_{j=1}^{M} \left(\mathbf{F}(\mathbf{x}_i^j) - y_j^i\right)^2 \\ &= \frac{1}{NM} \sum_{i=1}^{N} \sum_{j=1}^{M} \left(\frac{\mathbf{F}^{\mathrm{on}}(\mathbf{x}_i^j) + \mathbf{F}^{\mathrm{off}}(\mathbf{x}_i^j)}{2} - y_i^j\right)^2. \end{aligned} \tag{12}$$

## 4.2 Convergence Analysis

We employ NTK [35] to analyze the trajectory of networks $\mathbf{F}$ learned by Fed-CO$_2$ and networks $\mathbf{F}^{\mathrm{on}}$ learned by FedBN (the online model in Fed-CO$_2$). Existing studies [37, 38] have validated that the convergence rate of finite-width over-parameterized networks is controlled by the least eigenvalue of the induced kernel in the training process. Following the discovery in [38], we can decompose the NTK into a direction component $\mathbf{V}(t)/\alpha^2$ and a magnitude component $\mathbf{G}(t)$:

$$\frac{d\mathbf{F}}{dt} = -\boldsymbol{\Lambda}(t)(\mathbf{F}(t) - \mathbf{y}), \boldsymbol{\Lambda}(t) := \frac{\mathbf{V}(t)}{\alpha^2} + \mathbf{G}(t). \tag{13}$$

The specific forms of $\mathbf{V}(t)$ and $\mathbf{G}(t)$ are provided in the Appendix. Here, let $\lambda_{\min}(H)$ denote the minimum eigenvalue of matrix $H$. It is worth noting that both matrices $\mathbf{V}(t)$ and $\mathbf{G}(t)$ are positive semi-definite as they can be interpreted as covariance matrices. Based on this, we can deduce that $\lambda_{\min}(\boldsymbol{\Lambda}(t)) \geq \max\{\lambda_{\min}(\mathbf{V}(t)/\alpha^2), \lambda_{\min}(\mathbf{G}(t))\}$. From the NTK theory, the value of $\lambda_{\min}(\boldsymbol{\Lambda}(t))$ controls the convergence rate. Then, considering that $\alpha$ is the pre-defined parameter, for $\alpha \in (0, 1)$, convergence is dominated by $\mathbf{V}(t)$. Let $\boldsymbol{\Lambda}(t)$, $\boldsymbol{\Lambda}^{\mathrm{on}}(t)$ and $\boldsymbol{\Lambda}^{\mathrm{off}}(t)$ denote the evolution dynamics of Fed-CO$_2$, the online model and the offline model, respectively. Based on the work in [10], the auxiliary version of the Gram matrices $\mathbf{V}^\infty$, $\mathbf{V}_{\mathrm{on}}^\infty$, and $\mathbf{V}_{\mathrm{off}}^\infty$ for three models are strictly positive definite. Let the least eigenvalues $\lambda_{\min}(\mathbf{V}_{\mathrm{on}}^\infty) := \mu^{\mathrm{on}}$, $\lambda_{\min}(\mathbf{V}_{\mathrm{off}}^\infty) := \mu^{\mathrm{off}}$, and $\lambda_{\min}(\mathbf{V}^\infty) := \mu$, where $\mu^{\mathrm{on}}$, $\mu^{\mathrm{off}}$, and $\mu$ are all positive values. To be specific, we define Gram matrices $\mathbf{V}_{\mathrm{on}}^\infty$ and $\mathbf{V}_{\mathrm{off}}^\infty$ in Definition 4.2.

**Definition 4.2** *Given sample points $\{\mathbf{x}_p\}_{p=1}^{NM}$, we define the auxiliary Gram matrices $\mathbf{V}_{\mathrm{on}}^\infty \in \mathbb{R}^{NM \times NM}$ and $\mathbf{V}_{\mathrm{off}}^\infty \in \mathbb{R}^{NM \times NM}$ as*

$$\mathbf{V}_{\mathrm{on}_{pq}}^\infty := \mathbb{E}_{\mathbf{v} \sim \mathcal{N}(\mathbf{0}, \alpha^2 \mathbf{I})} (\alpha c)^2 \, \mathbf{x}_p^{\mathbf{v}^\perp} \mathbf{x}_q^{\mathbf{v}^\perp}, \quad \textit{(Online Model)} \tag{14}$$

$$\mathbf{V}_{\mathrm{off}_{pq}}^\infty := \mathbb{E}_{\mathbf{v} \sim \mathcal{N}(\mathbf{0}, \alpha^2 \mathbf{I})} (\alpha c)^2 \, \mathbf{x}_p^{\mathbf{v}^\perp} \mathbf{x}_q^{\mathbf{v}^\perp} \mathbb{1}\{i_p = i_q\}, \quad \textit{(Offline Model)}. \tag{15}$$

With the prerequisite provided in Appendix, the convergence performance of the online model, the offline model, and our Fed-CO$_2$ can be analyzed by comparing $\lambda_{\min}(\mathbf{V}_{\mathrm{on}}^\infty)$, $\lambda_{\min}(\mathbf{V}_{\mathrm{off}}^\infty)$, and $\lambda_{\min}(\mathbf{V}^\infty)$. Our main theoretical result is given in Theorem 4.3.

**Theorem 4.3** *For the V-dominated convergence, the convergence rate of Fed-CO$_2$ is faster than that of FedBN (the online model in Fed-CO$_2$).*

**Proof sketch** The core is to show $\lambda_{\min}(\mathbf{V}_{\text{on}}^{\infty}) \leq \lambda_{\min}(\mathbf{V}^{\infty})$. To prove the theorem, we first show that $\lambda_{\min}(\mathbf{V}_{\text{on}}^{\infty}) \leq \lambda_{\min}(\mathbf{V}_{\text{off}}^{\infty})$. Comparing Eq. 14 and 15, $\mathbf{V}_{\text{off}}^{\infty}$ takes the M $\times$ M block matrices on the diagonal of $\mathbf{V}_{\text{on}}^{\infty}$. Let $\mathbf{V}_i^{\infty}$ be the i-th M $\times$ M block matrices on the diagonal of $\mathbf{V}_{\text{on}}^{\infty}$. According to linear algebra, we have $\lambda_{\min}(\mathbf{V}_i^{\infty}) \geq \lambda_{\min}(\mathbf{V}_{\text{on}}^{\infty})$, for $i \in [N]$. With $\mathbf{V}_{\text{off}}^{\infty} = diag(\mathbf{V}_1^{\infty}, \cdots, \mathbf{V}_N^{\infty})$, it can be obtained that $\lambda_{\min}(\mathbf{V}_{\text{off}}^{\infty}) \geq \lambda_{\min}(\mathbf{V}_{\text{on}}^{\infty})$. Then, based on Eq. 10, 12, and 13, we can obtain $\lambda_{\min}(\mathbf{V}^{\infty}) = \lambda_{\min}(\frac{1}{2}(\mathbf{V}_{\text{on}}^{\infty} + \mathbf{V}_{\text{off}}^{\infty}))$. Therefore, we have $\lambda_{\min}(\mathbf{V}^{\infty}) \geq (\frac{1}{2}\lambda_{\min}(\mathbf{V}_{\text{on}}^{\infty}) + \frac{1}{2}\lambda_{\min}(\mathbf{V}_{\text{off}}^{\infty}))$. Since we have proven that $\lambda_{\min}(\mathbf{V}_{\text{off}}^{\infty}) \geq \lambda_{\min}(\mathbf{V}_{\text{on}}^{\infty})$, the result $\lambda_{\min}(\mathbf{V}^{\infty}) \geq \lambda_{\min}(\mathbf{V}_{\text{on}}^{\infty})$ can be achieved.

## 5 EXPERIMENTS

### 5.1 Experiment Settings

**Dataset and Data Heterogeneity.** We followed the footstep of prior research [39, 7, 5, 13] to study the label distribution skew heterogeneity with image datasets: CIFAR10 and CIFAR100 [40]. Specifically, we considered two different label distributions among participants: 1) Pathological Distribution, each client is randomly assigned 2 classes per client in CIFAR10 (10 classes per client in CIFAR100); 2) Dirichlet Distribution, each client gets its private data through partitioning of the datasets using a symmetric Dirichlet distribution with a default parameter $\alpha = 0.3$.

For feature skew heterogeneity, we conducted extensive experiments on three datasets: Digits, Office-Caltech10 [41], and DomainNet [42]. In this non-IID data setting, each domain serves as a client, with each client having access to all the data from its respective domain. Digits is composed of 5 different digit datasets with feature shift: SVHN [43], USPS [44], SynthDigits [45], MNIST-M [45] and MNIST [46]. Office-Caltech10 [41] owns data from four different domains including Amazon, DSLR, WebCam, and Caltech-256. DomainNet [42] is a challenging dataset that comprises six distinct domains, namely Clipart, Infograph, Painting, Quickdraw, Real, and Sketch. To explore the more realistic FL scenarios with both label distribution skew and feature skew, we exert a Dirichlet Distribution to datasets with feature shift. More details are supplemented in Appendix.

**Compared Benchmarks.** We compared our universal framework **Fed-CO$_2$** with fundamental algorithms including **SingleSet**, **FedAvg** [2] and **FedProx** [4], together with some refined FL algorithms: **FedPer** [11], **MOON** [7], **FedBN** [10], and **FedRoD** [12]. Here, **SingleSet** means each client trains their own model on their private data. We applied the linear version for **FedRoD**. For FL with feature skew issues, we further compared **COPA** [34], which was originally proposed to solve FDG and FDA problems. We now train the model on all domains in the dataset and evaluate its performance on each individual training client using the learned universal model.

### 5.2 Performance Evaluation

**Model Evaluation on Feature Skew.** Experiments are conducted on datasets Digits, Office-Caltech10, and DomainNet. Here, we present the results of experiments on Office-Caltech10, and DomainNet in Table 1 and Table 2. Full experiments are shown in Appendix. The performance of our Fed-CO$_2$ exhibits a dominant edge over state-of-the-art algorithms in all experiments, where the average accuracy on each domain increases by nearly $4\%$. The results confirm that our novel collaboration framework between online and offline models effectively addresses FL challenges associated with severe feature skew data heterogeneity.

Remarkably, when compared to algorithms FedBN [10] and COPA [34], which are explicitly designed to handle feature shift issues, our method consistently outperforms them across all sub-datasets. This phenomenon provides compelling evidence that our cooperation mechanism adapts to local feature shifts more effectively by leveraging both general domain-invariant and specialized domain-specific knowledge.

**Model Evaluation on Label Distribution Skew.** As shown in Table 3, our Fed-CO$_2$ outperforms a variety of state-of-the-art PFL algorithms in terms of average test accuracy across almost all cases in experiments with label distribution skew. Our state-of-the-art results across almost all cases

Table 1: Experiment results for FL with Feature Skew on Office-Caltech10.

| Methods | Office-Caltech10 | | | | |
| --- | --- | --- | --- | --- | --- |
| | Amazon | Caltech | DSLR | WebCam | Avg |
| SingleSet | 54.9±1.5 | 40.2±1.6 | 78.7±1.3 | 86.4±2.4 | 65.1±1.7 |
| FedAvg [2] | 54.1±1.1 | 44.8±1.0 | 66.9±1.5 | 85.1±2.9 | 62.7±1.6 |
| FedProx [4] | 54.2±2.5 | 44.5±0.5 | 65.0±3.6 | 84.4±1.7 | 62.0±2.1 |
| FedPer [11] | 49.0±1.2 | 37.1±2.4 | 57.7±3.7 | 79.7±2.1 | 56.0±1.1 |
| MOON [7] | 57.3±0.7 | 44.4±0.5 | 76.2±2.5 | 83.1±1.1 | 65.2±0.5 |
| FedRoD [12] | 60.4±2.3 | 45.3±0.9 | 73.7±2.5 | 83.7±2.3 | 65.8±1.4 |
| COPA [34] | 51.9±2.5 | 46.7±0.8 | 65.6±2.0 | 85.0±1.3 | 62.3±0.9 |
| FedBN [10] | 63.0±1.6 | 45.3±1.5 | 83.1±2.5 | 90.5±2.3 | 70.5±2.0 |
| Fed-$CO_2$ | **63.0±1.6** | **49.1±0.7** | **89.4±2.5** | **96.6±1.5** | **74.5±0.3** |

Table 2: Experiment results for FL with Feature Skew on DomainNet.

| Methods | DomainNet | | | | | | |
| --- | --- | --- | --- | --- | --- | --- | --- |
| | Clipart | Infograph | Painting | Quickdraw | Real | Sketch | Avg |
| SingleSet | 41.0±0.9 | 23.8±1.2 | 36.2±2.7 | 73.1±0.9 | 48.5±1.9 | 34.0±1.1 | 42.8±1.5 |
| FedAvg [2] | 48.8±1.9 | 24.9±0.7 | 36.5±1.1 | 56.1±1.6 | 46.3±1.4 | 36.6±2.5 | 41.5±1.5 |
| FedProx [4] | 48.9±0.8 | 24.9±1.0 | 36.6±1.8 | 54.4±3.1 | 47.8±0.8 | 36.9±2.1 | 41.6±1.6 |
| FedPer [11] | 40.4±0.8 | 25.7±0.6 | 37.3±0.6 | 62.5±1.2 | 47.4±0.5 | 32.8±0.8 | 41.0±0.3 |
| MOON [7] | 52.5±1.1 | 25.7±0.6 | 39.4±1.7 | 50.8±4.7 | 48.8±0.8 | 40.1±4.1 | 42.9±1.5 |
| FedRoD [12] | 50.8±1.6 | 26.3±0.2 | 40.1±1.8 | 66.8±1.8 | 51.5±1.1 | 39.1±2.0 | 45.7±0.7 |
| COPA [34] | 51.1±1.0 | 24.7±1.2 | 36.8±0.8 | 54.8±1.6 | 47.1±1.8 | 41.0±1.4 | 42.6±0.4 |
| FedBN [10] | 51.2±1.4 | 26.8±0.5 | 41.5±1.4 | 71.3±0.7 | 54.8±0.8 | 42.1±1.3 | 48.0±1.0 |
| Fed-$CO_2$ | **55.0±1.1** | **28.6±1.1** | **44.3±0.6** | **75.1±0.6** | **62.4±0.8** | **45.7±1.9** | **51.8±0.2** |

prove that our method can more effectively excavate and utilize local information to overcome data heterogeneity caused by label distribution imbalance than previous methods. It is worth highlighting that our Fed-$CO_2$ outperforms both SingleSet and FedBN, regardless of which one performs better individually. This observation confirms that Fed-$CO_2$ effectively integrates the general knowledge obtained by the online model and the specialized knowledge acquired by the offline model, resulting in enhanced performance.

Table 3: Experiment results for FL with Label Distribution Skew on CIFAR10 and CIFAR100. Experiments are conducted with two kinds of label distribution data heterogeneity: Pathological setting and Dirichlet setting.

| Methods | CIFAR10 | | CIFAR100 | |
| --- | --- | --- | --- | --- |
| | Pathological | Dirichlet | Pathological | Dirichlet |
| SingleSet | 85.85±0.05 | 68.38±0.06 | 49.54±0.05 | 21.39±0.05 |
| FedAvg [2] | 44.12±3.10 | 57.52±1.01 | 14.59±0.40 | 20.34±1.34 |
| FedProx [4] | 57.38±1.08 | 56.46±0.66 | 21.32±0.71 | 19.40±1.76 |
| FedPer [11] | 80.99±0.71 | 74.21±0.07 | 42.08±0.18 | 20.06±0.26 |
| MOON [7] | 48.43±3.18 | 54.49±1.87 | 17.89±0.76 | 19.73±0.71 |
| FedRoD [12] | **89.05±0.04** | 73.99±0.09 | 54.96±1.30 | 28.29±1.53 |
| FedBN [10] | 86.71±0.56 | 75.41±0.37 | 48.37±0.56 | 28.70±0.46 |
| Fed-$CO_2$ | 88.79±0.25 | **77.45±0.30** | **58.50±0.43** | **32.43±0.37** |

**Model Evaluation on Label Distribution Skew and Feature Skew.** Here, we evaluate our Fed-$CO_2$ and benchmark algorithms in FL scenarios with label distribution skew and feature skew. We exhibit the experiment results on Digits in Table 4, where we exerted a Dirichlet Distribution on each client with $\alpha = 0.3$. From the results, it is evident that, even in the presence of two types of data heterogeneity, our Fed-$CO_2$ outperforms various state-of-the-art algorithms with a significant margin in every sub-dataset. Therefore, we can conclude that under our universal cooperation framework, local clients not only make the most use of their local offline information but also benefit from the general knowledge shared by other clients for a better adaptation to local data, resulting in state-of-the-art performance in FL with both label distribution skew and feature skew. More experiments are provided in Appendix.

Table 4: Experiment results for FL with both Label Skew and Feature Skew on Digits.

| Methods | Digits | | | | | |
| --- | --- | --- | --- | --- | --- | --- |
| | MNIST | SVHN | USPS | SynthDigits | MNIST-M | Avg |
| SingleSet | 83.75±7.58 | 74.63±0.31 | 97.14±0.06 | 87.95±0.46 | 80.55±0.26 | 84.80±1.44 |
| FedAvg [2] | 89.27±6.39 | 57.23±5.43 | 94.60±1.05 | 81.30±2.51 | 71.71±7.59 | 78.82±4.03 |
| FedProx [4] | 87.09±7.83 | 53.40±7.38 | 90.55±7.47 | 78.40±7.05 | 69.98±6.55 | 75.88±6.90 |
| FedPer [11] | 96.92±0.02 | 72.86±0.11 | 97.09±0.12 | 87.82±0.02 | 83.69±0.07 | 87.68±0.02 |
| MOON [7] | 96.58±0.07 | 74.10±0.22 | 96.19±0.10 | 88.15±0.04 | 85.05±0.14 | 88.01±0.07 |
| FedRoD [12] | 96.65±0.06 | 77.05±0.16 | 96.88±0.13 | 89.59±0.06 | 86.18±0.11 | 89.27±0.05 |
| COPA [34] | 96.82±0.08 | 78.32±0.12 | 96.54±0.15 | 89.36±0.04 | 87.46±0.11 | 89.70±0.06 |
| FedBN [10] | 92.68±3.45 | 70.26±4.38 | 91.40±8.72 | 83.17±4.95 | 77.98±3.84 | 83.10±4.96 |
| Fed-CO$_2$ | **97.17±0.67** | **83.16±0.23** | **98.12±0.13** | **93.04±0.11** | **91.45±0.14** | **92.59±0.15** |

## 5.3 Component Analysis

In the presence of feature skew, we add intra-client and inter-client knowledge transfer mechanisms to foster the collaboration between the online and offline models for better local adaptation. Here, we evaluate the efficacy of these two knowledge transfer mechanisms on the challenging DomainNet dataset through three additional experiments, as presented in Table 5. In these experiments, we selectively removed specific mechanisms from Fed-CO$_2$ to investigate their individual contributions and functionalities.

According to the experimental results, we can observe that: 1) In the absence of intra-client and inter-client knowledge transfer mechanisms, Fed-CO$_2$ exhibits a significant decline in performance, resulting in a performance level comparable to that of benchmark methods. Therefore, cooperation relying solely on prediction fusion proves to be inadequate for FL with feature skew. 2) The removal of either the intra-client or the inter-client knowledge transfer mechanisms will also result in an average decline in performance. This phenomenon validates the effectiveness of our intra-client and inter-client knowledge transfer mechanisms, both of which contribute to enhancing the models' capability to adapt to local data distributions. 3) Compared with all three ablation experiments, our Fed-CO$_2$ consistently achieves the highest performance on average and across the majority of sub-datasets. These results firmly demonstrate that our cooperation framework effectively leverages both model-level and client-level collaboration through the intra-client knowledge transfer and inter-client knowledge transfer mechanisms, respectively, to adapt to feature shifts resulting from domain gaps. Further analysis of these two cooperation mechanisms is supplemented in the Appendix.

Table 5: Ablation study of Intra-client and Inter-client knowledge transfer mechanisms for FL with Feature Skew Data Heterogeneity on DomainNet.

| Methods | Clipart | Infograph | Painting | Quickdraw | Real | Sketch | Avg |
| --- | --- | --- | --- | --- | --- | --- | --- |
| Fed-CO$_2$ w/o Intra and Inter Transfer | 48.75±0.94 | 26.49±2.05 | 42.10±1.05 | 72.86±0.80 | 57.12±1.08 | 39.96±0.79 | 47.88±0.70 |
| Fed-CO$_2$ w/o Intra Transfer | 53.88±0.63 | 26.18±0.71 | 42.94±0.92 | **75.10±0.28** | 61.94±0.70 | **46.68±0.75** | 51.12±0.19 |
| Fed-CO$_2$ w/o Inter Transfer | 50.42±0.72 | 26.97±1.07 | 43.94±0.69 | 74.14±0.64 | 58.14±0.85 | 42.02±0.80 | 49.27±0.46 |
| Fed-CO$_2$ | **55.02±1.13** | **28.58±1.10** | **44.27±0.62** | 75.08±0.62 | **62.37±0.76** | 45.67±1.95 | **51.83±0.25** |

## 6 CONCLUSIONS

In this work, we proposed a new universal FL framework called Fed-CO$_2$ that is capable of handling label distribution skew and feature skew even when both are present in the same data. The core of our approach is to foster cooperation between the online and offline models, leveraging the benefits of online general knowledge and offline specialized knowledge to effectively adapt to local data distribution. To improve model collaboration in FL with feature shifts, we designed two novel knowledge transfer mechanisms: one intra-client and the other inter-client. These mechanisms facilitate mutual learning between the online and offline models, concurrently enhancing the model's capacity to generalize across different domains. Comparisons with a wide range of state-of-the-art methods on five benchmark datasets consistently show that Fed-CO$_2$ yields superior performance in addressing both label distribution skew and feature skew challenges, both individually and in combination. Extending Fed-CO$_2$ to scenarios with noisy training data is under consideration in our future work.

## Acknowledgement

This work was supported by NSFC (No.62303319), Shanghai Sailing Program (21YF1429400, 22YF1428800), Shanghai Local College Capacity Building Program (23010503100), Shanghai Frontiers Science Center of Human-centered Artificial Intelligence (ShangHAI), MoE Key Laboratory of Intelligent Perception and Human-Machine Collaboration (ShanghaiTech University), and Shanghai Engineering Research Center of Intelligent Vision and Imaging.

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

# A   Dataset Setup Details

The details of non-IID settings of mentioned datasets are presented in the section. Our two settings of label distribution skew, namely the Pathological setting and the Dirichlet setting are similar to preceding works [5, 13]. In the Pathological setting, where each client $i$ is randomly assigned two (ten) classes from CIFAR10 (CIFAR100), we first sampled $q_c^i \sim U(0.4, 0.6)$ for the selected class $c$ and then calculated the sample rate for this class $c$ on client $i$ with $sr_c^i = \frac{q_c^i}{\sum_j q_c^j}$. In the Dirichlet setting, we partitioned the datasets randomly utilizing a symmetric Dirichlet distribution with default parameter $\alpha = 0.3$. Specifically, for CIFAR100, we leveraged its coarse labels following previous works [5, 39]. In detail, a two-stage Pachinko allocation method was adopted to partition samples in CIFAR100. For each client, this method first generates a Dirichlet distribution with default parameter $\alpha = 0.3$ over the coarse labels and then generates another Dirichlet distribution with parameter $\beta = 10$ over the corresponding fine labels. The local training batch size was set to 64. In both non-IID partitions, the classes and their distribution in each client's training and test sets remain consistent.

We adopted the setting used in FedBN [10] for feature skew heterogeneity, which involved adjusting the local training batch size to 32 to maintain consistency. Additionally, we followed the implementation of selecting only the top 10 classes based on data amount from DomainNet to create the dataset for our experiments. For experiments with label distribution skew and feature skew, we partitioned each sub-dataset into several parts with a Dirichlet Distribution and selected one part as the new training set. Then we adjusted the label distribution in the test set to the same distribution in the new training set. Specifically, based on the number of training images in each dataset, we divided each sub-dataset in Digits into three parts, each sub-dataset in Office-Caltech10 into two parts, and each sub-dataset in DomainNet into five parts. Table 6 summarizes the datasets and the number of clients.

Table 6: Datasets Statistics.

| Dataset | Label Distribution Skew | Feature Skew | Client Number |
|---|---|---|---|
| CIFAR10 | ✓ | | 100 |
| CIFAR100 | ✓ | | 100 |
| Digits | | ✓ | 5 |
| Office-Caltech10 | | ✓ | 4 |
| DomainNet | | ✓ | 6 |

# B   Implementation Details

**Network Modules.** Similar to prior works [47, 48], we adopted a ConvNet [46] with two convolutional layers and three fully-connected layers for methods in experiments conducted under label distribution skew setting. With regards to feature skew setting, experiments were performed with a backbone similar to FedBN [10]. In other words, for the Digits dataset, we applied a ConvNet [46] with four convolutional layers and three fc layers. For Office-Caltech10 and DomainNet dataset, we applied one Alexnet [49] as the default model backbone.

In our experimental evaluations of model performance under label distribution skew heterogeneity, we added an additional BN layer after each convolutional layer and fully connected (fc) layer, except for the last layer to algorithms: FedBN [10] and Fed-CO$_2$. Meanwhile, in experiments pertaining to feature skew heterogeneity, as our default network incorporated a BN layer after each convolutional and fc layer (except for the last layer), we omitted the BNs in our classifiers for methods other than FedBN [10] and Fed-CO$_2$. All other Convnet network architectures used for the Digits dataset, as well as the Alexnet architecture employed for Office-Caltech10 and DomainNet, are identical to those used in FedBN [10].

**Concrete Implementation.** For experiments evaluating model performance on label distribution skew, we set up 100 local clients and randomly sampled 5% of them to participate in each communication round. We trained every algorithm with 1500 communication rounds. Meanwhile, in experiments about feature skew non-iid heterogeneity, all clients participated in each communication round and every algorithm was trained with 300 communication rounds. All training tasks were optimized with an SGD optimizer with the default learning rate 0.01. We used PyTorch [50] to implement our algorithms with an RTX 2080 Ti GPU.

For experiments on CIFAR10 and CIFAR100, we tested the model performance every 5 rounds during its last 200 communication rounds. The average and variance of test accuracy were reported. For all experiments with feature skew, we run 5 times and reported its mean and variance.

**Evaluation Metrics.** We evaluated the learned personal models on the private test data of each client. We examined the model's performance on each domain using the same method as FedBN [10] in FL with feature skew. Furthermore, for experiments on label distribution skew, we calculated the accuracy using the same protocol as FedTP [13].

## C Convergence

In this section, we first provide the preliminary and complete theorem proof. Then, we empirically demonstrate that our Fed-CO$_2$ shows faster and more robust convergence performance than FedBN and other benchmark methods on FL issues with Feature Skew.

### C.1 Preliminary

Following the V-dominated convergence rate for FedAvg (Theorem C.1) in [38], we can derive the convergence rate for the online model, the offline model, and Fed-CO$_2$ Corollary C.2.

**Theorem C.1 ($V$-dominated convergence for FedAvg [38])** *Suppose the network is initialized as in Eq. 11 with $\alpha \leq 1$, trained using gradient descent and Assumption 4.1 holds. Assuming the loss function used for training the neural network is the square loss, and the target values $\mathbf{y}$ satisfy $\|\mathbf{y}\|_\infty = O(1)$. If $m = \Omega(N^4 M^4 log(NM/\delta)/\mu_0^4)$, then with probability $1 - \delta$,*

1. *for iterations $t = 0, 1, \cdots$, the evolution matrix $\mathbf{\Lambda}(t)$ satisfies $\lambda_{\min}(\mathbf{\Lambda}(t)) \geq \frac{\mu_0}{2\alpha^2}$;*

2. *training with gradient descent of step-size $\eta = O\left(\frac{\alpha^2}{\|\mathbf{V}^\infty\|_2}\right)$ converges linearly as*

$$\|\mathbf{f}(t) - \mathbf{y}\|_2^2 \leq \left(1 - \frac{\eta\mu_0}{2\alpha^2}\right)^t \|\mathbf{f}(0) - \mathbf{y}\|_2^2.$$

**Corollary C.2** *Under the same assumption in Theorem C.1, with probability $1 - \delta$, for iterations $t = 0, 1, \cdots$, we have*

1. *The evolution matrix $\mathbf{\Lambda}^{\mathrm{on}}(t)$ satisfies $\lambda_{\min}(\mathbf{\Lambda}^{\mathrm{on}}(t)) \geq \frac{\mu^{\mathrm{on}}}{2\alpha^2}$ and training with gradient descent of step-size $\eta = O\left(\frac{\alpha^2}{\|\mathbf{V}^\infty_{\mathrm{on}}\|_2}\right)$ converges linearly as $\|\mathbf{F}^{\mathrm{on}}(t) - \mathbf{y}\|_2^2 \leq \left(1 - \frac{\eta\mu^{\mathrm{on}}}{2\alpha^2}\right)^t \|\mathbf{F}^{\mathrm{on}}(0) - \mathbf{y}\|_2^2.$*

2. *The evolution matrix $\mathbf{\Lambda}^{\mathrm{off}}(t)$ satisfies $\lambda_{\min}(\mathbf{\Lambda}^{\mathrm{off}}(t)) \geq \frac{\mu^{\mathrm{off}}}{2\alpha^2}$ and training with gradient descent of step-size $\eta = O\left(\frac{\alpha^2}{\|\mathbf{V}^\infty_{\mathrm{off}}\|_2}\right)$ converges linearly as $\|\mathbf{F}^{\mathrm{off}}(t) - \mathbf{y}\|_2^2 \leq \left(1 - \frac{\eta\mu^{\mathrm{off}}}{2\alpha^2}\right)^t \|\mathbf{F}^{\mathrm{off}}(0) - \mathbf{y}\|_2^2.$*

3. *The evolution matrix $\mathbf{\Lambda}(t)$ satisfies $\lambda_{\min}(\mathbf{\Lambda}(t)) \geq \frac{\mu}{2\alpha^2}$ and training with gradient descent of step-size $\eta = O\left(\frac{\alpha^2}{\|\mathbf{V}^\infty\|_2}\right)$ converges linearly as $\|\mathbf{F}(t) - \mathbf{y}\|_2^2 \leq \left(1 - \frac{\eta\mu}{2\alpha^2}\right)^t \|\mathbf{F}(0) - \mathbf{y}\|_2^2.$*

Therefore, the exponential factor of convergence for the online model, the offline model, and Fed-CO$_2$ are controlled by the smallest eigenvalue of $\mathbf{V}^{\mathrm{on}}(t)$, $\mathbf{V}^{\mathrm{off}}(t)$, and $\mathbf{V}(t)$, respectively. The convergence performance of the online model, the offline model, and our Fed-CO$_2$ can be analyzed by comparing $\lambda_{\min}(\mathbf{V}^\infty_{\mathrm{on}})$, $\lambda_{\min}(\mathbf{V}^\infty_{\mathrm{off}})$, and $\lambda_{\min}(\mathbf{V}^\infty)$.

### C.2 Theorem Proof

To prove Theorem 4.3, we will first prove SingleSet (the offline model in Fed-CO$_2$) converges faster than FedBN (the online model in Fed-CO$_2$). Gram matrix $\mathbf{V}^{\mathrm{on}}(t)$ and $\mathbf{G}^{\mathrm{on}}(t)$ of FedBN can be

obtained from work [10] as:

$$\mathbf{V}_{pq}^{\text{on}}(t) = \frac{1}{m} \sum_{k=1}^{m} (\alpha c_k^{\text{on}})^2 \gamma_{k,i_p}^{\text{on}}(t) \gamma_{k,i_q}^{\text{on}}(t) \left\| \mathbf{v}_k^{\text{on}}(t) \right\|_{\mathbf{S}_{i_p}}^{-1} \left\| \mathbf{v}_k^{\text{on}}(t) \right\|_{\mathbf{S}_{i_q}}^{-1} \left\langle \mathbf{x}_p^{\mathbf{v}_k^{\text{on}\,i_p}(t)^\perp}, \mathbf{x}_q^{\mathbf{v}_k^{\text{on}\,i_q}(t)^\perp} \right\rangle \tag{16}$$
$$\mathbb{1}_{pk}(t)\mathbb{1}_{qk}(t),$$

$$\mathbf{G}_{pq}^{\text{on}}(t) = \frac{1}{m} \sum_{k=1}^{m} c_k^{\text{on}^2} \left\| \mathbf{v}_k^{\text{on}}(t) \right\|_{\mathbf{S}_{i_p}}^{-1} \left\| \mathbf{v}_k^{\text{on}}(t) \right\|_{\mathbf{S}_{i_q}}^{-1} \sigma\left(\mathbf{v}_k^{\text{on}}(t)^\top \mathbf{x}_p\right) \sigma\left(\mathbf{v}_k^{\text{on}}(t)^\top \mathbf{x}_q\right) \mathbb{1}\{i_p = i_q\}, \tag{17}$$

where $\mathbb{1}_{pk}(t) := \mathbb{1}_{\{\mathbf{v}_k(t)^\top \mathbf{x}_p \geq 0\}}$.

Similarly, the Gram matrix $\mathbf{V}^{\text{off}}(t)$ for method SingleSet with $\mathbf{F}^{\text{off}}$ can be computed as:

$$\mathbf{V}_{pq}^{\text{off}}(t) = \frac{1}{m} \sum_{k=1}^{m} \alpha^2 c_{k,i_p}^{\text{off}} c_{k,i_q}^{\text{off}} \gamma_{k,i_p}^{\text{off}}(t) \gamma_{k,i_q}^{\text{off}}(t) \left\| \mathbf{v}_{k,i_p}^{\text{off}}(t) \right\|_{\mathbf{S}_{i_p}}^{-1} \left\| \mathbf{v}_{k,i_q}^{\text{off}}(t) \right\|_{\mathbf{S}_{i_q}}^{-1}$$
$$\left\langle \mathbf{x}_p^{\mathbf{v}_{k,i_p}^{\text{off}\,i_p}(t)^\perp}, \mathbf{x}_q^{\mathbf{v}_{k,i_q}^{\text{off}\,i_q}(t)^\perp} \right\rangle \mathbb{1}_{pk}(t)\mathbb{1}_{qk}(t)\mathbb{1}\{i_p = i_q\}. \tag{18}$$

To compare the convergence rates of SingleSet and FedBN, we compare the exponential factor in the convergence rates, which are $(1 - \frac{\eta\mu^{\text{off}}}{2\alpha^2})$ and $(1 - \frac{\eta\mu^{\text{on}}}{2\alpha^2})$, respectively. Then, considering that $\alpha$ is the pre-defined parameter, it reduces to comparing $\mu^{\text{on}} = \lambda_{min}(\mathbf{V}_{\text{on}}^\infty)$ and $\mu^{\text{off}} = \lambda_{min}(\mathbf{V}_{\text{off}}^\infty)$. Comparing Eq. 16 and 18, $\mathbf{V}_{\text{off}}^\infty$ takes the M × M block matrices on the diagonal of $\mathbf{V}_{\text{on}}^\infty$:

$$\mathbf{V}_{\text{on}}^\infty = \begin{bmatrix} \mathbf{V}_1^\infty & \mathbf{V}_{1,2}^\infty & \cdots & \mathbf{V}_{1,N}^\infty \\ \mathbf{V}_{1,2}^\infty & \mathbf{V}_2^\infty & \cdots & \mathbf{V}_{2,N}^\infty \\ \vdots & \vdots & \ddots & \vdots \\ \mathbf{V}_{1,N}^\infty & \mathbf{V}_{2,N}^\infty & \cdots & \mathbf{V}_N^\infty \end{bmatrix}, \quad \mathbf{V}_{\text{off}}^\infty = \begin{bmatrix} \mathbf{V}_1^\infty & 0 & \cdots & 0 \\ 0 & \mathbf{V}_2^\infty & \cdots & 0 \\ \vdots & \vdots & \ddots & \vdots \\ 0 & 0 & \cdots & \mathbf{V}_N^\infty \end{bmatrix},$$

where $\mathbf{V}_i^\infty$ is the i-th $M \times M$ block matrices on the diagonal of $\mathbf{V}_{\text{on}}^\infty$. Therefore, we have:

$$\lambda_{\min}(\mathbf{V}_i^\infty) \geq \lambda_{\min}(\mathbf{V}_{\text{on}}^\infty), \quad \forall i \in [N].$$

Since the eigenvalues of $\mathbf{V}_{\text{off}}^\infty$ are exactly the union of eigenvalues of $\mathbf{V}_i^\infty$, we get:

$$\lambda_{\min}(\mathbf{V}_{\text{off}}^\infty) = \min_{i \in [N]} \{\lambda_{\min}(\mathbf{V}_i^\infty)\},$$
$$\geq \lambda_{\min}(\mathbf{V}_{\text{on}}^\infty).$$

Hence, $(1 - \frac{\eta\mu^{\text{on}}}{2\alpha^2}) \geq (1 - \frac{\eta\mu^{\text{off}}}{2\alpha^2})$ and we get the first-stage conclusion that the convergence rate of method SingleSet is faster than that in FedBN.

Then we go to prove Fed-CO$_2$ is faster than FedBN, as well. With Eq. 10, 12, and 13, we get:

$$\frac{d\mathbf{F}}{dt} = \frac{1}{2}\left(\frac{d\mathbf{F}^{\text{on}}}{dt} + \frac{d\mathbf{F}^{\text{off}}}{dt}\right)$$
$$= -\left(\frac{1}{2}\mathbf{\Lambda}^{\text{on}}(t) + \frac{1}{2}\mathbf{\Lambda}^{\text{off}}(t)\right)\left(\frac{1}{2}(\mathbf{F}^{\text{on}}(t) + \mathbf{F}^{\text{off}}(t)) - \mathbf{y}\right) \tag{19}$$
$$= -\left(\frac{1}{2}\mathbf{\Lambda}^{\text{on}}(t) + \frac{1}{2}\mathbf{\Lambda}^{\text{off}}(t)\right)(\mathbf{F}(t) - \mathbf{y}).$$

Based on Eq. 19, we get the following relationships among three models:

$$\mathbf{\Lambda}(t) := \frac{1}{2}\left(\mathbf{\Lambda}^{\text{on}}(t) + \mathbf{\Lambda}^{\text{off}}(t)\right)$$
$$= \frac{1}{2}\left(\frac{\mathbf{V}^{\text{on}}(t)}{\alpha^2} + \mathbf{G}^{\text{on}}(t)\right) + \frac{1}{2}\left(\frac{\mathbf{V}^{\text{off}}(t)}{\alpha^2} + \mathbf{G}^{\text{off}}(t)\right) \tag{20}$$
$$= \frac{1}{2}\left(\frac{\mathbf{V}^{\text{on}}(t) + \mathbf{V}^{\text{off}}(t)}{\alpha^2}\right) + \frac{1}{2}\left(\mathbf{G}^{\text{on}}(t) + \mathbf{G}^{\text{off}}(t)\right).$$

To compare convergence rates of Fed-CO$_2$ and FedBN, we need to compare their exponential factor $(1 - \frac{\eta\mu}{2\alpha^2})$ and $(1 - \frac{\eta\mu^{\mathrm{on}}}{2\alpha^2})$, as well. Similar to the former proof, the problem reduces to compare $\mu = \lambda_{\min}\left(\frac{1}{2}\left(\mathbf{V}_{\mathrm{on}}^\infty + \mathbf{V}_{\mathrm{off}}^\infty\right)\right)$ with $\mu^{\mathrm{on}} = \lambda_{\min}(\mathbf{V}_{\mathrm{on}}^\infty)$. With previous proof result $\lambda_{\min}(\mathbf{V}_{\mathrm{off}}^\infty) \geq \lambda_{\min}(\mathbf{V}_{\mathrm{on}}^\infty)$, we get

$$\lambda_{\min}\left(\frac{1}{2}\left(\mathbf{V}_{\mathrm{on}}^\infty + \mathbf{V}_{\mathrm{off}}^\infty\right)\right) \geq \frac{1}{2}\lambda_{\min}(\mathbf{V}_{\mathrm{on}}^\infty) + \frac{1}{2}\lambda_{\min}(\mathbf{V}_{\mathrm{off}}^\infty)$$
$$\geq \frac{1}{2}\lambda_{\min}(\mathbf{V}_{\mathrm{on}}^\infty) + \frac{1}{2}\lambda_{\min}(\mathbf{V}_{\mathrm{on}}^\infty)$$
$$= \lambda_{\min}(\mathbf{V}_{\mathrm{on}}^\infty).$$

Therefore, we have $(1 - \frac{\eta\mu^{\mathrm{on}}}{2\alpha^2}) \geq (1 - \frac{\eta\mu}{2\alpha^2})$. This theoretical conclusion guarantees a faster convergence in our Fed-CO$_2$ than FedBN.

### C.3 Experimental Performance

Here, we empirically analyze the convergence of test accuracy for various algorithms, as well as the convergence behavior of both the online and offline models. The results for sub-dataset WebCam and SVHN are illustrated in Fig. 2 as examples. As Fig. 2 exhibits, among various FL algorithms, our Fed-CO$_2$ achieves the highest accuracy with significantly faster convergence speed and demonstrates much more robust behavior. Compared with its online and offline models, Fed-CO$_2$ consistently surpasses both models, exhibiting a smoother curve, which proves the success in fusing domain-invariant and domain-specific knowledge.

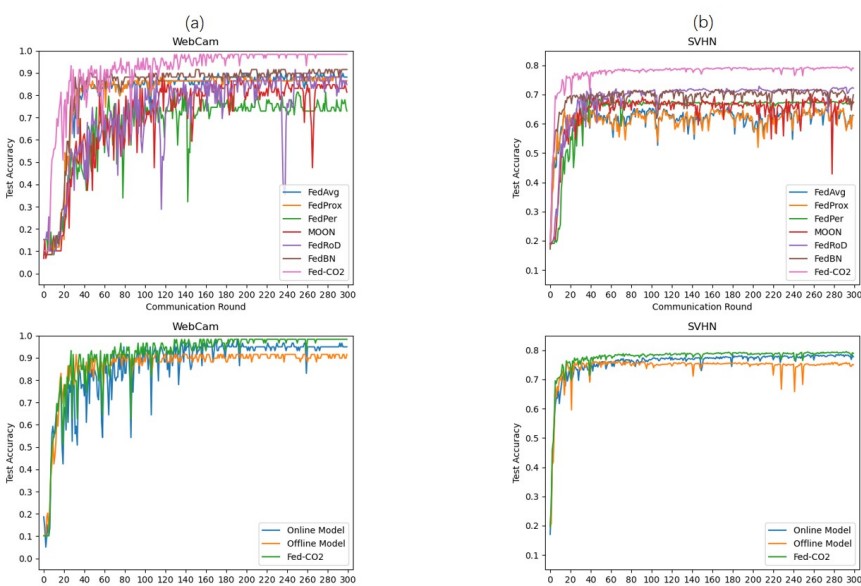

Figure 2: Convergence of test accuracy on sub-datasets. Fig. 2(a) and Fig. 2(b) illustrate convergence behavior on WebCam from Office-Caltech10 and SVHN from Digits, respectively. In each sub-graph, we exhibit convergence performance comparison among different algorithms (Top) and comparison with online and offline models (Bottom). Fed-CO$_2$ exhibits faster and more robust convergence.

## D  Fed-CO$_2$ Algorithm

Here, we provide a detailed description of the algorithm for our universal FL framework, Fed-CO$_2$. The learning process for FL with and without Feature Skew is presented in Algorithm 1 and Algorithm 2, respectively.

**Algorithm 1:** Fed-CO$_2$ for FL with Feature Skew

---

**Input:** $T-$ number of communication rounds, $N-$ number of clients.

1   Set parameters $\theta_{g,0}^{\text{on}}$, $\theta_{p,i,0}^{\text{on}}$, $\theta_{i,0}^{\text{off}}$;

2   **for** *each communication rounds* $t \in \{1,...,T\}$ **do**

3     **Communicate** $\{\overline{\phi}_j^{\text{off}}\}_{j=1}^N$, $\widetilde{\theta}_{g,t-1}^{\text{on}}$ to all clients;

4     **for** *each client* $i \in [1, N]$ **do**

5       **Initialize** $\theta_{i,t}^{\text{on}} = \{\widetilde{\theta}_{g,t-1}^{\text{on}}, \theta_{p,i,t-1}^{\text{on}}\}$, $\theta_{i,t}^{\text{off}} = \theta_{i,t-1}^{\text{off}}$, $\bar{\theta}_i^{\text{on}} = \theta_{i,t}^{\text{on}}$, $\bar{\theta}_i^{\text{off}} = \theta_{i,t}^{\text{off}}$;

6       $(\theta_{i,t}^{\text{on}}, \theta_{i,t}^{\text{off}}) \leftarrow$ **Intra-client Knowledge Transfer**$(\bar{\theta}_i^{\text{off}}, \theta_{i,t}^{\text{on}}, \bar{\theta}_i^{\text{on}}, \theta_{i,t}^{\text{off}})$ [Eq. 4 and Eq. 5];

7       $(\theta_{i,t}^{\text{on}}, \theta_{i,t}^{\text{off}}) \leftarrow$ **Inter-client Knowledge Transfer**$(\theta_{i,t}^{\text{on}}, \theta_{i,t}^{\text{off}}, \{\overline{\phi}_j^{\text{off}}\}_{j=1}^N)$ [Eq. 7];

8       **Communicate** $\theta_{g,i,t}^{\text{on}}$, $\phi_{i,t}^{\text{off}}$ to the server

9     **end**

10    **Aggregate** $\widetilde{\theta}_{g,t}^{\text{on}} = \frac{1}{N}\sum_{i=1}^N \theta_{g,i,t}^{\text{on}}$ [Eq. 2];

11    **Construct** $\{\overline{\phi}_j^{\text{off}}\}_{j=1}^N = \{\phi_{i,t}^{\text{off}}\}_{i=1}^N$;

12 **end**

13 **return** $\widetilde{\theta}_{g,t}^{\text{on}}$, $\theta_{p,i,t}^{\text{on}}$ and $\theta_{i,t}^{\text{off}}$

---

**Algorithm 2:** Fed-CO$_2$ for FL without Feature Skew

---

**Input:** $T-$ number of communication rounds, $N-$ number of clients.

1   Set parameters $\theta_{g,0}^{\text{on}}$, $\theta_{p,i,0}^{\text{on}}$, $\theta_{i,0}^{\text{off}}$;

2   **for** *each communication rounds* $t \in \{1,...,T\}$ **do**

3     **Communicate** $\widetilde{\theta}_{g,t-1}^{\text{on}}$ to all clients;

4     **for** *each client* $i \in [1, N]$ **do**

5       **Initialize** $\theta_{i,t}^{\text{on}} = \{\widetilde{\theta}_{g,t-1}^{\text{on}}, \theta_{p,i,t-1}^{\text{on}}\}$, $\theta_{i,t}^{\text{off}} = \theta_{i,t-1}^{\text{off}}$;

6       $(\theta_{i,t}^{\text{on}}, \theta_{i,t}^{\text{off}}) \leftarrow$ **Local Training**$(\theta_{i,t}^{\text{on}}, \theta_{i,t}^{\text{off}})$;

7       **Communicate** $\theta_{g,i,t}^{\text{on}}$ to the server

8     **end**

9    **Aggregate** $\widetilde{\theta}_{g,t}^{\text{on}} = \frac{1}{N}\sum_{i=1}^N \theta_{g,i,t}^{\text{on}}$ [Eq. 2];

10 **end**

11 **return** $\widetilde{\theta}_{g,t}^{\text{on}}$, $\theta_{p,i,t}^{\text{on}}$ and $\theta_{i,t}^{\text{off}}$

---

Table 7: Experiment results for FL with Feature Skew on Digits.

| Methods | Digits | | | | | |
|---|---|---|---|---|---|---|
| | MNIST | SVHN | USPS | SynthDigits | MNIST-M | Avg |
| SingleSet | 94.38±0.07 | 65.25±1.07 | 95.16±0.12 | 80.31±0.38 | 77.77±0.47 | 82.00±0.40 |
| FedAvg [2] | 95.87±0.20 | 62.86±1.49 | 95.56±0.27 | 82.27±0.44 | 76.85±0.54 | 82.70±0.60 |
| FedProx [4] | 95.75±0.21 | 63.08±1.62 | 95.58±0.31 | 82.34±0.37 | 76.64±0.55 | 82.70±0.60 |
| FedPer [11] | 96.21±0.02 | 67.61±0.04 | 96.53±0.02 | 83.88±0.02 | 81.89±0.03 | 85.22±0.01 |
| MOON [7] | 96.25±0.04 | 65.48±0.49 | 95.05±0.12 | 82.89±0.15 | 80.57±0.23 | 84.05±0.16 |
| FedRoD [12] | 96.09±0.08 | 71.50±0.25 | 96.42±0.08 | 85.51±0.04 | 81.79±0.09 | 86.26±0.07 |
| COPA [34] | 96.27±0.05 | 72.90±0.17 | 95.99±0.06 | 85.52±0.04 | 83.08±0.22 | 86.75±0.02 |
| FedBN [10] | 96.57±0.13 | 71.04±0.31 | 96.97±0.32 | 83.19±0.42 | 78.33±0.66 | 85.20±0.40 |
| Fed-CO$_2$ | **97.66±0.07** | **77.56±0.60** | **97.78±0.13** | **88.68±0.08** | **87.84±0.12** | **89.91±0.11** |

# E  More Experimental Results On Benchmark Datasets

Here, we supplement more experimental results on benchmark datasets. Firstly, we present the experimental results on Digits dataset for FL with feature skew in Table 7. As the table shows, our Fed-$CO_2$ surpasses other various state-of-the-art algorithms with a substantial margin, which further proves the effectiveness of our framework in adapting local feature shifts.

Then, we show experimental results in FL with both label distribution skew and feature skew on DomainNet and Office-Caltech10 in Table 8 and Table 9. We applied a Dirichlet Distribution with $\alpha = 2.0$ to DomainNet and another Dirichlet Distribution with $\alpha = 1.0$ to Office-Caltech10. It can be observed that when facing two severe forms of data heterogeneity, algorithms demonstrated significantly lower performance on Office-Caltech10 and DomainNet compared to FL scenarios with feature skew alone. Even in such a challenging task, our Fed-$CO_2$ still achieves the highest average accuracy and outperforms locally-trained models. This result proves that Fed-$CO_2$ succeeds in making use of both general domain-invariant and specialized domain-specific knowledge to adapt to extreme local data heterogeneity.

Table 8: Experiment results for FL with both Label Skew and Feature Skew on DomainNet.

| Methods | DomainNet | | | | | | |
|---|---|---|---|---|---|---|---|
| | Clipart | Infograph | Painting | Quickdraw | Real | Sketch | Avg |
| SingleSet | 16.3±2.5 | 17.4±1.5 | 18.8±3.7 | 12.6±1.1 | 12.3±1.6 | 12.4±4.3 | 15.0±0.6 |
| FedAvg [2] | 9.4±1.7 | 9.2±1.4 | 15.2±1.3 | 10.0±1.7 | 9.1±0.7 | 10.2±1.4 | 10.5±0.3 |
| FedProx [4] | 9.6±1.4 | 10.8±1.6 | 12.2±1.2 | 10.1±0.7 | 10.3±1.5 | 11.0±2.7 | 10.7±1.1 |
| FedPer [11] | 16.6±1.2 | 15.3±1.3 | 18.6±2.4 | **14.4±1.0** | 11.9±1.7 | **15.3±2.9** | 15.4±0.5 |
| FedRoD [12] | 13.6±1.1 | 15.4±2.1 | 15.9±1.8 | 10.6±2.3 | 11.8±1.0 | 11.4±1.4 | 13.1±0.6 |
| COPA [34] | 8.4±0.9 | 10.2±2.5 | 13.2±2.5 | 10.2±1.9 | 11.1±1.3 | 10.9±1.3 | 10.7±0.9 |
| FedBN [10] | 11.0±1.1 | 12.9±1.3 | 13.0±2.0 | 11.8±0.6 | 10.4±1.4 | 11.0±2.2 | 11.7±0.4 |
| Fed-$CO_2$ | **18.8±2.4** | **17.9±1.8** | **19.5±2.5** | 14.1±1.1 | **12.2±2.2** | 14.6±4.4 | **16.2±0.5** |

Table 9: Experiment results for FL with both Label Skew and Feature Skew on Office-Caltech10.

| Methods | Office-Caltech10 | | | | |
|---|---|---|---|---|---|
| | Amazon | Caltech | DSLR | WebCam | Avg |
| SingleSet | 14.6±2.3 | 15.2±0.3 | 10.0±1.2 | 3.4±0.0 | 10.8±0.6 |
| FedAvg [2] | 15.9±1.7 | 15.0±2.4 | 11.2±1.5 | 3.0±2.7 | 11.3±0.7 |
| FedProx [4] | 14.0±1.3 | 13.0±1.0 | 9.4±2.0 | 5.1±2.4 | 10.4±0.8 |
| FedPer [11] | 16.5±0.6 | 16.1±0.3 | 11.9±1.2 | 1.7±0.0 | 11.5±0.4 |
| FedRoD [12] | 14.7±0.9 | 12.1±1.0 | 6.25±0.0 | **6.4±0.7** | 9.9±0.5 |
| COPA [34] | 17.5±1.7 | 11.5±0.7 | 8.1±2.5 | 4.7±2.7 | 10.4±1.0 |
| FedBN [10] | **19.5±2.5** | 12.6±1.9 | 11.9±2.3 | 2.4±0.8 | 11.6±0.4 |
| Fed-$CO_2$ | 17.5±0.9 | **16.6±1.2** | **12.5±0.0** | 3.7±1.3 | **12.6±0.5** |

Additionally, to explore broader real-life data heterogeneity issues with both feature skew and label distribution skew, we conducted further experiments by exerting different levels of label distribution skew on the Digits dataset. Specifically, we partitioned each sub-dataset of Digits using the Dirichlet Distribution, with different values of $\alpha \in \{0.3, 0.7, 1.0, 1.5, 2.0\}$. A smaller $\alpha$ value indicates a more severe label distribution imbalance. The experimental results are demonstrated in Fig. 3. It is evident that in every instance of label distribution imbalance, our Fed-$CO_2$ exhibits a significant advantage over other benchmark methods in its sub-dataset as well as the overall average performance. Therefore, we can firmly conclude that our universal FL framework, Fed-$CO_2$, effectively addresses a wide range of real-life data heterogeneity issues.

# F  Additional Experiments and Analyses

## F.1  Effectiveness of Cooperation Framework

The experimental results of testing accuracy on five benchmark datasets have validated that our Fed-$CO_2$ effectively combines online general knowledge and offline specialized knowledge, resulting in improved adaptation to local data distribution in FL scenarios with label imbalance and feature

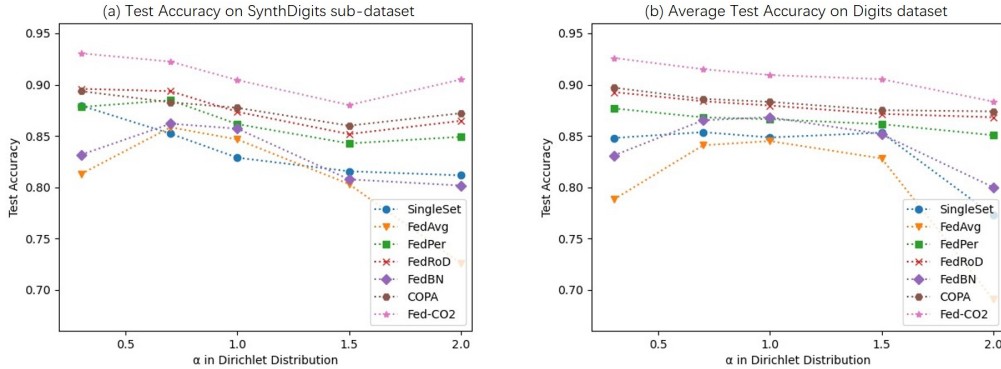

Figure 3: The test accuracy of our Fed-CO$_2$ and benchmark algorithms in Federated Learning with Feature Skew and varying levels of Label Distribution Skew. Fig. 3(a) depicts the model performance on the SynthDigits sub-dataset, while Fig. 3(b) demonstrates the average model performance across all sub-datasets in Digits.

shifts. To further confirm the effectiveness of the cooperation between the online and offline models, we conducted additional experiments for FL with label imbalance and feature shifts in the following parts.

**Online and Offline Models in FL with Label Distribution Skew.** For FL with label imbalance, we split the CIFAR10 dataset for 20 clients with various types and degrees of label distribution skew. In the Pathological case, each client $i$ was randomly assigned two classes following the strategies mentioned in Appendix A. In the Dirichlet case, we partitioned the dataset randomly utilizing the Dirichlet distribution with different parameters $\alpha \in \{0.3, 0.7, 1.0\}$ to evaluate model performances under various extents of label distribution imbalance. In these experiments, we compared the model trained by our Fed-CO$_2$ with its online and offline models.

The experimental results have been exhibited in Fig. 4. It is evident that the online model outperforms the offline model when the label distribution imbalance is moderate, while the offline model excels when the imbalance is severe. Across all extensive experiments, Fed-CO$_2$ consistently surpasses both the standalone online model and the standalone offline model in terms of test accuracy statistics, exhibiting higher average and median values. At the client level, Fed-CO$_2$ improves prediction accuracy for the majority of local clients. Although Fed-CO$_2$ may not surpass the superior model between the online and offline models on a limited number of clients, it consistently outperforms the weaker model and achieves performance levels approaching that of the superior model. Therefore, we can conclude that our collaborative FL framework, Fed-CO$_2$, which combines the strengths of online and offline models, is exceptionally effective. It successfully integrates online general knowledge and offline specialized knowledge, resulting in an improved adaptation to local data distribution under various cases.

**Online and Offline Models in FL with Feature Skew.** For FL with feature skew, we conducted experiments that evaluate the performance of the online and offline models on DomainNet. Based on the experimental results shown in Table 10, we have the following observations: First, the online model outperforms the offline model on most clients, except for the Quickdraw client. This phenomenon validates our hypothesis that when the feature shift is severe, the model aggregation will lead to the loss of important local offline information, resulting in the model's failure to adapt to such significant data heterogeneity. Second, prediction fusion is even inferior to the online and offline models for some clients. This result reflects that prediction fusion is not sufficient to fuse online general knowledge and offline specialized knowledge in FL with severe data heterogeneity issues. Therefore, for FL with feature skew, intra-client and inter-client knowledge transfer mechanisms are required to boost model and client cooperation under our novel framework.

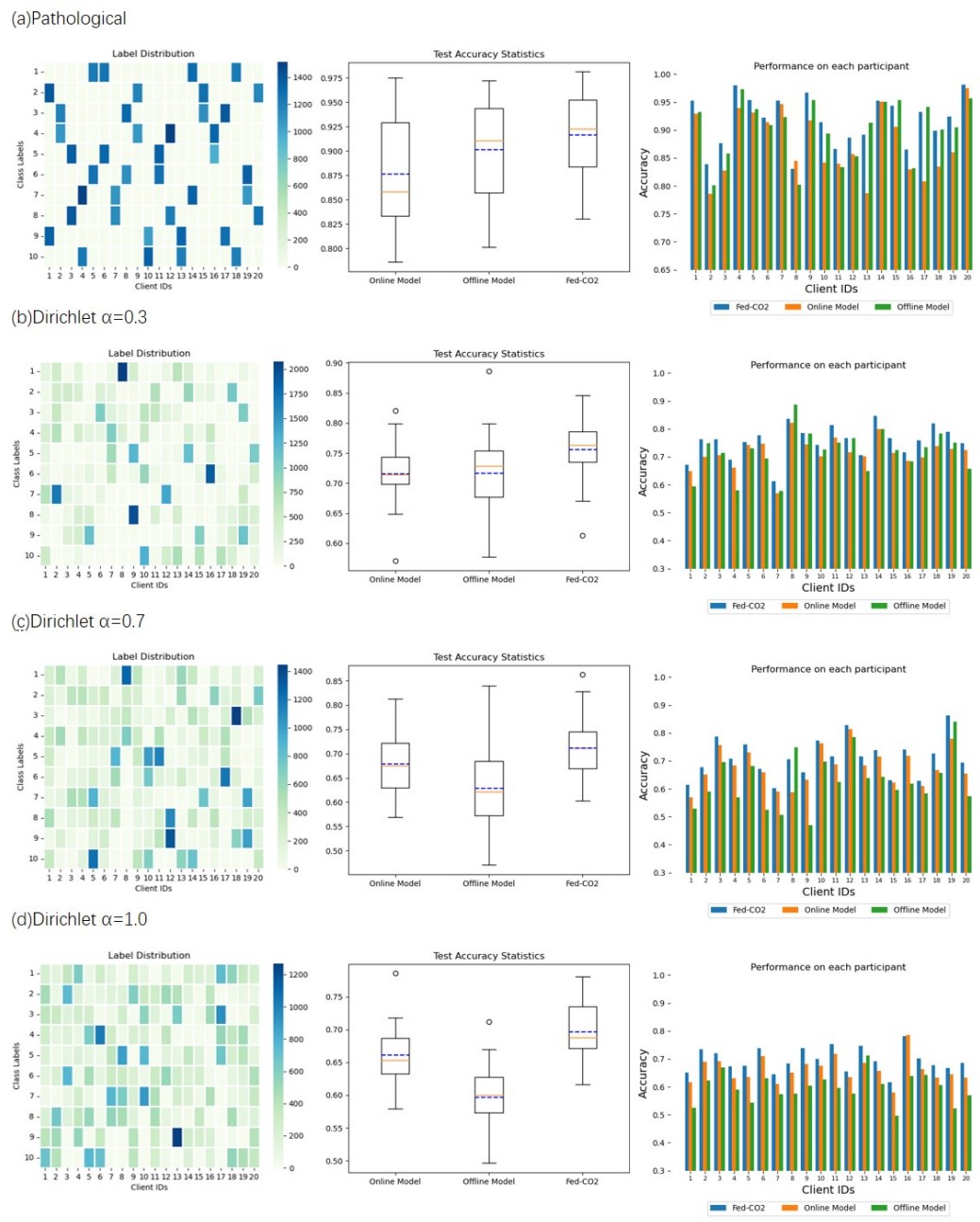

Figure 4: Comparison results among the online model, the offline model, and our Fed-CO$_2$ with different kinds of label distribution skew. In each subgraph, we illustrate label distribution among 20 clients (**Left**), the classification test accuracy statistics (**Middle**), where the full line represents the median accuracy and the dotted line represents the average accuracy among the 20 clients, and test accuracy in each client (**Right**).

Table 10: Performance of Online and Offline Models in FL with Feature Skew on DomainNet

| Methods | Clipart | Infograph | Painting | Quickdraw | Real | Sketch | Avg |
|---|---|---|---|---|---|---|---|
| Online Model | 51.2±1.4 | 26.8±0.5 | 41.5±1.4 | 71.3±0.7 | 54.8±0.8 | 42.1±1.3 | 48.0±1.0 |
| Offline Model | 41.0±0.9 | 23.8±1.2 | 36.2±2.7 | 73.1±0.9 | 48.5±1.9 | 34.0±1.1 | 42.8±1.5 |
| Fed-CO$_2$ (no intra and inter knowledge transfer) | 48.7±0.9 | 26.5±2.0 | 42.1±1.0 | 72.9±0.8 | 57.1±1.1 | 40.0±0.8 | 47.9±0.7 |
| Fed-CO$_2$ | **55.0±1.1** | **28.6±1.1** | **44.3±0.6** | **75.1±0.6** | **62.4±0.8** | **45.7±1.9** | **51.8±0.2** |

## F.2  Knowledge Transfer Mechanisms in Feature Skew

As mentioned in the main paper, intra-client and inter-client knowledge transfer mechanisms foster model-level and client-level collaboration to better make use of general domain-invariant and specialized domain-specific knowledge for a better adaptation to local data distribution. In this section, we did further ablation studies on the challenging DomainNet dataset to investigate two mechanisms from a fine-grained angle.

**Ablation Study in Intra-client Knowledge Transfer Mechanism.** In this section, we investigated the impact of the intra-client knowledge transfer mechanism on the online and offline models by excluding the inter-client knowledge transfer mechanism from Fed-CO$_2$. Specifically, we performed separate intra-client knowledge transfers: from the offline model to the online model and from the online model to the offline model. The results are demonstrated in Table 11 and we can discover that knowledge transfer is beneficial to both online and offline models. By transferring specialized domain-specific knowledge to the online model, we can observe an improvement in the performance of the Quickdraw client, which has a more extreme non-IID data distribution and relies more on local offline information. Conversely, when transferring general domain-invariant knowledge to the offline model, we could notice enhanced performance in the remaining clients that present milder data heterogeneity and require more global information. Hence, by incorporating the full intra-client knowledge transfer mechanism, our Fed-CO$_2$ effectively utilizes both online domain-invariant and offline domain-specific knowledge, resulting in improved average test accuracy.

Table 11: Ablation Study of Intra-client Knowledge Transfer mechanism on DomainNet

| Method | Clipart | Infograph | Painting | Quickdraw | Real | Sketch | Avg |
|---|---|---|---|---|---|---|---|
| No intra transfer | 48.7±0.9 | 26.5±2.0 | 42.1±1.0 | 72.9±0.8 | 57.1±1.1 | 40.0±0.8 | 47.9±0.7 |
| Intra transfer to online | 48.1±1.2 | 26.2±1.3 | 42.1±1.2 | 74.0±0.5 | 57.0±1.0 | 40.4±1.0 | 48.0±0.3 |
| Intra transfer to offline | **50.9±0.7** | 26.6±0.7 | 42.1±1.1 | 72.8±0.5 | **59.1±0.3** | 41.3±1.8 | 48.8±0.2 |
| Full intra transfer | 50.4±0.7 | **27.0±1.1** | **43.9±0.7** | **74.1±0.6** | 58.1±0.8 | **42.0±0.8** | **49.3±0.5** |

**Ablation Study in Inter-client Knowledge Transfer Mechanism.** In this part, we removed the intra-client knowledge transfer mechanism from Fed-CO$_2$ and did experiments to investigate the impact of inter-client knowledge transfer mechanism on online and offline models. The experiments included exerting inter-client knowledge transfer mechanism on the online model, the offline model, and both models. Table 12 presents the experimental results. The results demonstrate that knowledge from other domains benefits both the online and offline models, with the online model showing more remarkable improvement. This observation aligns consistently with the purpose of the online model, which focuses on acquiring general domain-invariant knowledge that can be applied across various domains. Furthermore, the application of the inter-client knowledge transfer mechanism to both the online and offline models yields optimal performance, which means simultaneously enhancing domain generalization ability for online and offline models can exploit domain-invariant knowledge more effectively. In summary, our inter-client knowledge transfer mechanism enables each local client to acquire and exploit more domain-invariant knowledge resulting in a better adaptation to local data distribution.

Table 12: Ablation Study of Inter-client Knowledge Transfer Mechanism on DomainNet

| Methods | Clipart | Infograph | Painting | Quickdraw | Real | Sketch | Avg |
|---|---|---|---|---|---|---|---|
| No inter transfer | 48.7±0.9 | 26.5±2.0 | 42.1±1.0 | 72.9±0.8 | 57.1±1.1 | 40.0±0.8 | 47.9±0.7 |
| Online model with inter transfer | 51.3±0.7 | **27.1±1.0** | **44.2±1.3** | 74.1±0.7 | 61.4±0.7 | 43.5±1.5 | 50.3±0.5 |
| Offline model with inter transfer | 51.8±1.3 | 26.3±1.8 | 43.3±1.3 | 74.2±1.0 | 59.0±2.2 | 43.1±1.6 | 49.6±0.6 |
| Full inter transfer | **53.9±0.6** | 26.2±0.7 | 42.9±0.9 | **75.1±0.3** | **61.9±0.7** | **46.7±0.7** | **51.1±0.2** |

## F.3  Data Heterogeneity of Noise

Noise is another common factor causing data heterogeneity issues in real FL scenarios, where each client has the same label distribution but is subject to various degrees of noise. Therefore, we implemented additional experiments on this new scenario to make the initial attempt. In detail, we divided the dataset CIFAR10 into 10 clients in a way to make them independently and identically distributed. Then we applied the increasing level of random Gaussian noise with mean $\mu_i = 0$ and standard deviation $\sigma_i = \frac{\sigma_M}{N-1} * i$, where $i \in \{0, 1, \cdots, N-1\}$. In this series of initial experiments,

we let the client number and the sample rate be 0.5 with $\sigma_M \in \{0, 5, 10, 15, 20\}$. The backbone of each method is the same as that in other experiments.

As presented in Fig. 5, the experimental results reveal that our Fed-$CO_2$ consistently exhibits better and more robust performance across nearly all experiments, particularly in scenarios with severe noise. These experiments will also serve as a source of inspiration for our future work.

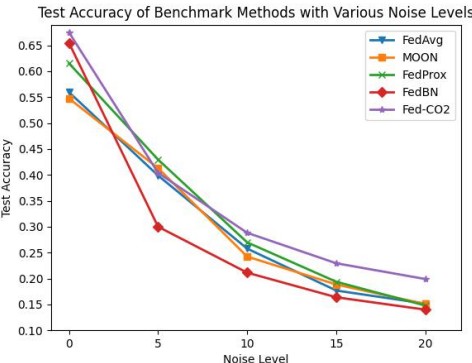

Figure 5: Test accuracy on dataset CIFAR10 for Fed-$CO_2$ and some benchmark methods with different degrees of data heterogeneity caused by noise.

### F.4 Effects of Data Number and Client Number

The sample size of the training set and the client number are significant factors. Therefore, in this section, we conducted a series of experiments to explore their influence.

To investigate the effect of the training sample number, we utilized dataset Digits and set 10% of its training data as the original training dataset. Then, we selected a certain ratio $\gamma$ of the original training dataset as a new training dataset for each client, where $\gamma \in \{0.2, 0.4, 0.6, 1.0, 2.0\}$. The results shown in Fig. 6 demonstrate that our Fed-$CO_2$ have an edge over various benchmark methods in almost every case, especially when the number of training data is very limited.

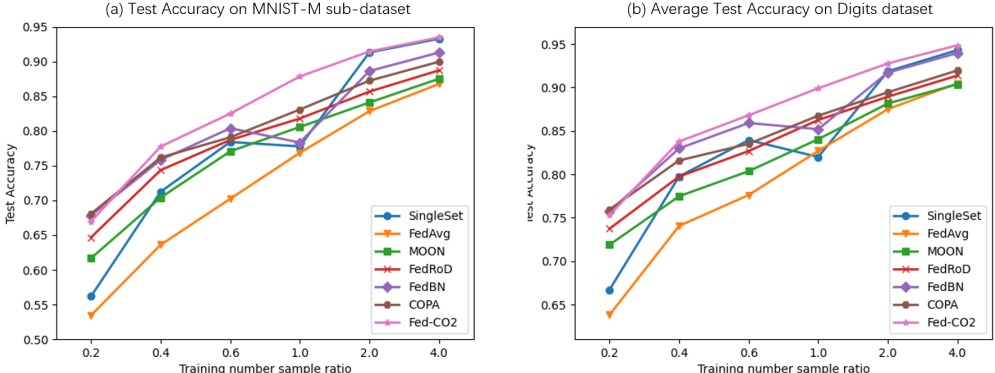

Figure 6: The effect of the training number ratio on Fed-$CO_2$ and benchmark algorithms for Federated Learning with Feature Skew. Fig. 6(a) depicts the model performance on the MNIST-M sub-dataset, while Fig. 6(b) demonstrates the average model performance across all sub-datasets in Digits.

As for the effect of client number, we employed dataset CIFAR10 with Dirichlet setting ($\alpha = 0.3$) to conduct a series of experiments on benchmark methods with client number $N \in \{50, 100, 150, 300, 500\}$. The experimental results shown in Table 13 demonstrate that our Fed-$CO_2$ consistently outperforms a bunch of SOTA methods in every case even if the client number is substantial. The results in Table 13 and Fig. 6 prove that our Fed-$CO_2$ owns higher scalability than benchmark methods.

Table 13: The influence of client numbers on CIFAR10

| #client | 50 | 100 | 150 | 300 | 500 |
|---|---|---|---|---|---|
| SingleSet | 71.28±0.08 | 68.69±0.07 | 70.05±0.06 | 64.62±0.04 | 60.31±0.07 |
| FedAvg [2] | 41.20±3.13 | 52.85±2.30 | 33.34±11.15 | 42.63±3.85 | 34.31±5.68 |
| FedProx [4] | 45.35±6.94 | 36.00±4.42 | 34.53±6.59 | 37.44±6.23 | 34.58±5.18 |
| MOON [7] | 40.25±3.31 | 52.38±2.16 | 34.93±9.74 | 49.58±3.93 | 37.35±5.80 |
| FedRoD [12] | 65.18±1.14 | 66.08±0.92 | 67.00±1.43 | 65.46±1.50 | 64.68±4.30 |
| FedBN [10] | 74.40±0.61 | 75.43±0.39 | 78.38±0.44 | 76.06±0.46 | 69.24±1.21 |
| Fed-CO$_2$ | **77.14±0.37** | **77.21±0.29** | **80.06±0.25** | **78.53±0.31** | **75.59±0.50** |

## F.5  Personalized Parts in the Online model

In order to enhance the adaptation to local data distribution, it is necessary to personalize certain critical parameters in the online model to preserve the essential offline specialized knowledge. In this section, we investigated and analyzed which specific parts of our online model should be personalized for optimal performance in FL with feature skew. The most common strategy is to personalize Batch Normalization layers or the Classifier Head. Therefore in our Fed-CO$_2$ approach, we conducted three experiments with different personalized components in the online model to compare and determine the optimal strategy. The results for three datasets, namely DomainNet, Office-Caltech10, and Digits, are illustrated in Fig. 7. It is evident from the observations that personalizing the Classification Head has a negative impact on almost every sub-dataset, resulting in lower classification accuracy. Based on the results, we can conclude that personalizing Batch Normalization layers in our online model is the best strategy to adapt to feature shifts.

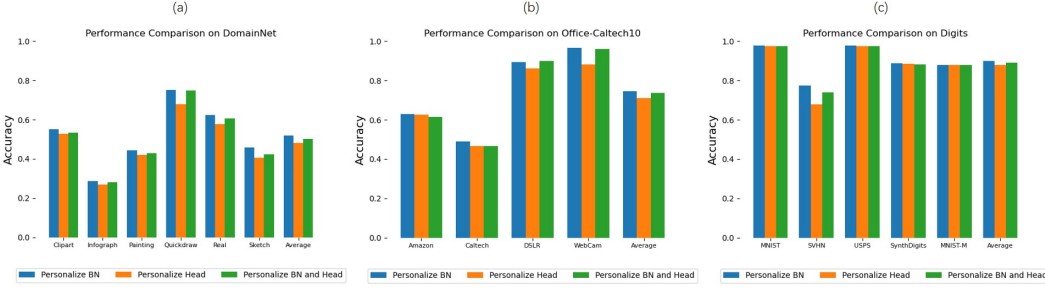

Figure 7: The test accuracy for our Fed-CO$_2$ with different personalized parts in the online model on DomainNet, Office-Caltech10, and Digits.

