# OpenReview forum: "Fed-CO$_{2}$: Cooperation of Online and Offline Models for Severe Data Heterogeneity in Federated Learning"
_NeurIPS.cc/2023/Conference — NeurIPS 2023 poster_

### Official Review · Reviewer_NQAC · 2023-07-05

**Soundness:** 4 excellent
**Presentation:** 3 good
**Contribution:** 4 excellent
**Rating:** 6
**Confidence:** 2

**Summary:**

In this paper, the authors focus on tackling the feature skew issue, and introduces a new approach that utilizes knowledge distillation. Specifically, the proposed method transfers the knowledge between the FL online model and the local offline model to enable them to have both domain invariant and domain specific knowledge.

**Strengths:**

Strength:

1. The proposed approach of mutual learning between the offline and online models is novel.

2. Theoretical analysis of the proposed method is provided (though they are all in the appendix)

3. Experimental results on multiple datasets with both label distribution skew and feature skew demonstrate promising outcomes.

**Weaknesses:**

1. Lack of important baselines:

	a. Ditto: Since Ditto is a very classic pFL method, it should be included as the baseline.

	b. FedMask [1]: There is no hypernetwork based method that directly personalizing parts of the model parameters. FedMask is one of the SOTA method, which learns distinct binary masks for the last several layers of the local models, and aggregates the masks with an intersection operation.

2. Lack of some experiments:

	The proposed method actually contains three components, intra-client, inter-client knowledge distillation, and the fuse of online, offline models. In the ablation study, the authors focus on the experiment of the previous two parts, and the fuse part is only mentioned in the label distribution skew setting (Figure 4 in the appendix). How about conducting experiments to evaluate the performance of the online and offline models in the feature distribution skew setting, i.e., the DomainNet dataset?


**Questions:**

1. Refer weakness 1 and weakness 2.

2. Regarding the order of two knowledge transfers:  Why is Intra-Client Knowledge Transfer executed before Inter-Client Knowledge Transfer？

3. Why is the theoretical analysis part only included in the appendix? In my opinion, at least the main theorem should be presented in the main body of the paper, while the assumptions, lemmas, and proofs could be included in the appendix.

---

> ### Author Rebuttal · Authors · 2023-08-09
>
> >**Q1**: Lack of important baselines:
> a. Ditto: Since Ditto is a very classic pFL method, it should be included as the baseline.
> b. FedMask [1]: There is no hypernetwork based method that directly personalizing parts of the model parameters. FedMask is one of the SOTA method, which learns distinct binary masks for the last several layers of the local models, and aggregates the masks with an intersection operation.
>
> **R1**: Thank you for providing us with two important baselines.  We will cite them in Related Work of our final version but will not compare our method with them. Generally, it is hard to compare all the existing methods in federated learning.
>
> For Ditto, one of our baseline methods FedRoD [12] has shown higher performance than it in the case of label distribution skew, while our Fed-CO$_2$ outperforms FedRoD[12] on FL with label distribution skew, feature skew, and both. This can clearly prove our method is better than Ditto on PFL issues. For FedMask, this algorithm is designed to address label distribution skew issues with no special design to overcome domain gaps. Moreover, we have not yet found published code online. It is hard to reimplement this algorithm.
>
> >**Q2**: Lack of some experiments:
> The proposed method actually contains three components, intra-client, inter-client knowledge distillation, and the fuse of online, offline models. In the ablation study, the authors focus on the experiment of the previous two parts, and the fuse part is only mentioned in the label distribution skew setting (Figure 4 in the appendix). How about conducting experiments to evaluate the performance of the online and offline models in the feature distribution skew setting, i.e., the DomainNet dataset?
>
> **R2**: Here we supplement experiments that evaluate the performance of the online and offline models in FL with the feature skew on DomainNet. The experimental results and relevant analyses will be added in the Appendix of our final version.
>
> Based on the experimental results shown in the Table below, we have the following observations:
> First, the online model outperforms the offline model on most clients, except for the Quickdraw client. This phenomenon validates our hypothesis that when the feature shift is severe, the model aggregation will lead to the loss of important local offline information, resulting in the model's failure to adapt to such significant data heterogeneity. Second, prediction fusion is even inferior to the online and offline models for some clients. This result reflects that prediction fusion is not sufficient to fuse online general knowledge and offline specialized knowledge.
>
> Therefore, for FL with feature skew, intra-client and inter-client knowledge transfer mechanisms are required to boost model and client cooperation under our novel framework.
>
>
> | Method                                        | Clipart  | Infograph | Painting | Quickdraw | Real     | Sketch   | Avg     |
> | -------- | -------- | -------- | -------- | -------- | -------- | -------- | -------- |
> | Online Model                                  | 51.2±1.4 | 26.8±0.5  | 41.5±1.4 | 71.3±0.7  | 54.8±0.8 | 42.1±1.3 | 48.0±1.0 |
> |Offline Model                                 | 41.0±0.9 | 23.8±1.2  | 36.2±2.7 | 73.1±0.9  | 48.5±1.9 | 34.0±1.1 | 42.8±1.5 |
> |Fed-CO2 (no intra and inter knowledge transfer) | 48.7±0.9 | 26.5±2.0  | 42.1±1.0 | 72.9±0.8  | 57.1±1.1 | 40.0±0.8 | 47.9±0.7 |
> |Fed-CO2                                       | 55.0±1.1 | 28.6±1.1  | 44.3±0.6 | 75.1±0.6  | 62.4±0.8 | 45.7±1.9 | 51.8±0.2
>
> >**Q3**: Regarding the order of two knowledge transfers: Why is Intra-Client Knowledge Transfer executed before Inter-Client Knowledge Transfer?
>
> **R3**: If the Inter-Client Knowledge Transfer is executed before the Intra-Client Knowledge Transfer, the global general information in the online model will be damaged and the initial model for the local training process will be weaker.
>
> The online model gets global general information after parameter aggregation on the server. But parameter aggregation will also cause the online model to forget some local specialized information. Meanwhile, the offline model keeps all learned local specialized information but cannot access any global information.
>
> Intra-Client Knowledge Transfer not only aids the online model in compensating for lost local information but also equips the offline model with some global information. This approach initializes the offline and online models better for the next local training.
>
> Inter-client Knowledge Transfer utilizes local data to not only adapt to local data distribution but also concurrently enhance the domain generalization ability for the online and offline models during the local training process. When the Inter-Client Knowledge Transfer is executed ahead of the Intra-Client Knowledge Transfer, the global general knowledge in the online model will unavoidably be covered by the learned local knowledge. In addition, the initial online model will lose local knowledge and the initial offline model will lose global knowledge in this case.
>
> >**Q4**: Why is the theoretical analysis part only included in the appendix? In my opinion, at least the main theorem should be presented in the main body of the paper, while the assumptions, lemmas, and proofs could be included in the appendix.
>
> **R4**: We appreciate your valuable advice and will present the Main Theorem in the main body of the paper in our final version.

---

> > ### Comment · Reviewer_NQAC · 2023-08-15
> >
> > Thanks for your response! All my concerns have been addresses, and I will keep my score.

---

### Official Review · Reviewer_y4gh · 2023-07-09

**Soundness:** 2 fair
**Presentation:** 2 fair
**Contribution:** 2 fair
**Rating:** 6
**Confidence:** 3

**Summary:**

The paper proposes a personalized federated learning (PFL) method to handle both label and feature distribution skew.Specifically, each clients holds a partially perosnalized model (personalization happens at the level of batch-normaizliation layers), and a fully personalized local model.  The paper conducts a series of numerical sumilations to evaluate the performance of the proposed method and to compares with state-of-the-art PFL methods.

**Strengths:**

- The paper is overall well-written and easy to follow.
- The proposed method shows promising performance and outperforms competitors in most cases.
- The proposed method does not induce any communication overhead compared to standard federated aggregation operations since the fully personalized offline models are not uploaded to the server.

**Weaknesses:**

- The paper does not provide stong explantion to motivate the proposed method. In particular, it is unclear where (6) and (7) come from, and why they take this specific form.
- The proposed method induces a memory overhead in comparison to  standard federated averaging.
- Apart from proposing a PFL with good performance, the paper does not provide any insights helping to understand personalization.
- The paper does not provide any theoretical guarantees.
- I think that (3) is not accurate. Maybe a division by 2 in the RHS of (3) is missing.

**Questions:**

- Could you please provide more intuition on the particular choice of the loss function in (6) and (7)?
- Could you please double check (3)?

**Limitations:**

- The proposed method induces a memory overhead in comparison to  standard federated averaging.
- The paper does not provide any theoretical guarantees.

---

> ### Author Rebuttal · Authors · 2023-08-09
>
> >**Q1**: The paper does not provide stong explantion to motivate the proposed method. In particular, it is unclear where (6) and (7) come from, and why they take this specific form.
>
> **R1**: The motivation of our proposed method is answered together with Q3. Formulas (6) and (7) describe the generalization loss function $L_{gen}$ for the online and offline models separately, which force feature extractor $f_{i}$ to produce robust and well-generalized features that can be recognized by frozen classifiers $\overline{C}\_{j}^{\rm off} (j\neq i)$ introduced from other clients' offline models.
>
> This function facilitates the acquisition of domain-invariant knowledge for the online and offline models from other clients. Our design is inspired by the belief that if the feature extractor owns general knowledge from other clients then the image features extracted by it should be well recognized by classifiers trained on other clients.
>
> Take formula (6) as an example, which transfers general knowledge from other clients to the online model. $f_{i}^{\rm on}(\eta_{i}^{\rm on}; x_{k})$ represents image features extracted by the feature extractor of the online model. Then we give these features to frozen classifiers from other clients. $\overline{C}\_{j}^{\rm off}(\overline{\phi}\_{j}^{\rm off}; f\_{i}^{\rm on}(\eta\_{i}^{\rm on}; x\_{k}))$ represents the prediction logits provided by the classifier from client j.
>
> >**Q2**: The proposed method induces a memory overhead in comparison to standard federated averaging.
>
> **R2**: Yes, we do need more memory overhead but our method achieves SOTA in cases with severe data heterogeneity, where standard federated averaging shows a quite poor performance. In addition, some published methods induced a memory overhead (even more than ours) as well. For example, KNN-Per [51] employed a local memory bank to save local image features; PerFCL [52] kept an extra local feature extractor to obtain local information; [12] utilized an extra personalized classifier or hyper network to deal with heterogeneity issues.
>
> [51] Marfoq, Othmane, et al. "Personalized federated learning through local memorization." International Conference on Machine Learning. PMLR, 2022.
>
> [52] Zhang, Yupei, et al. "Doubly contrastive representation learning for federated image recognition." Pattern Recognition 139 (2023): 109507.
>
> >**Q3**: Apart from proposing a PFL with good performance, the paper does not provide any insights helping to understand personalization.
>
> **R3**: The key of Fed-CO$_2$ relies on the fusion of offline specialized knowledge and online general knowledge that is crucial for federated personalization. Our insight comes from the observation (as mentioned in line 40) that in certain instances of extreme data heterogeneity, models trained by existing personalized algorithms may exhibit inferior performance compared to the locally-trained model. Conversely, in FL scenarios with milder heterogeneity, partially personalized models trained by PFL algorithms perform better.
>
> Based on this observation, we raised the question (as mentioned in line 46): Is there a more effective approach to fuse the online general knowledge and the offline specialized knowledge to achieve better performance? To answer this question, we conducted a series of experiments and designed our effective universal framework Fed-CO$_2$.
>
> >**Q4**: The paper does not provide any theoretical guarantees.
>
> **R4**: Actually, we have provided a detailed theoretical analysis from the perspective of Neural Tangent Kernel [47] in Appendix C. As reviewer NQAC suggests in Q4, we will present the main theorem in the main body of the paper in our final version.
>
> >**Q5**: I think that (3) is not accurate. Maybe a division by 2 in the RHS of (3) is missing.
>
> **R5**: In fact, whether (3) is divided by 2 or not remains consistent for the final prediction result. This is because Formula (3) is only utilized during the testing phase for class prediction. In our framework, the online model $F^{\rm on}$ and offline model $F^{\rm off}$ are trained separately. We will add this illustration in the final version.

---

### Official Review · Reviewer_5E9k · 2023-07-10

**Soundness:** 3 good
**Presentation:** 3 good
**Contribution:** 3 good
**Rating:** 5
**Confidence:** 2

**Summary:**

The paper presents Fed-CO2, a novel framework for Federated Learning (FL), a distributed learning paradigm that allows multiple clients to collectively learn a global model without sharing their private data. FL's effectiveness depends heavily on the quality of the data used for training. Data heterogeneity issues, like label distribution skew and feature skew, can significantly impact FL's performance. Traditionally, most studies have focused on dealing with label distribution skew, while a few recent ones have started addressing feature skew. These forms of data heterogeneity have been studied separately and have not been integrated within a unified FL framework.

Fed-CO2 aims to address this gap by developing a framework that handles both label distribution skew and feature skew. The framework utilizes a cooperation mechanism between the Online and Offline models: the online model learns general knowledge shared among all clients, and the offline model is trained locally to learn each individual client's specialized knowledge.  To further enhance model cooperation in the presence of feature shifts, the authors design an intra-client knowledge transfer mechanism to reinforce mutual learning between the online and offline models. Additionally, they introduce an inter-client knowledge transfer mechanism to enhance the models' domain generalization ability. Through extensive experiments, Fed-CO2 outperforms a wide range of existing personalized federated learning algorithms, handling label distribution skew and feature skew, both individually and collectively.

**Strengths:**

- The cooperation mechanism between the online and offline models is a novel and effective way to deal with data heterogeneity issues. This mechanism allows for general learning shared among all clients, as well as specialized learning unique to each client.
- The Fed-CO2 framework represents a comprehensive approach to addressing data heterogeneity in Federated Learning. It doesn't just focus on label distribution skew or feature skew, but rather addresses both these challenges within a unified framework.
- The extensive experiments show that the Fed-CO2 model performs better than a wide range of current personalized federated learning algorithms. This indicates the effectiveness and potential of the Fed-CO2 model.

**Weaknesses:**

- The model may be complex to implement, especially in environments where the number of clients is very large, due to the dual model structure (online and offline) and the knowledge transfer mechanisms.
- Despite being designed to handle heterogeneity, the performance of the framework is still reliant on the quality of the data used for training, which may not always be guaranteed, especially in a federated setting.
- The paper does not clearly articulate how the proposed method scales with increased data, feature dimensions, or numbers of clients. Understanding the scalability of the method is crucial in real-world applications.

**Questions:**

- What are the computational costs associated with the proposed intra-client and inter-client knowledge transfer mechanisms?
- How does Fed-CO2 handle potential security and privacy concerns associated with inter-client knowledge transfer?
- How does Fed-CO2 handle scenarios where the label distribution skew and feature skew are extreme?
- Can the framework handle other types of data heterogeneity beyond label distribution skew and feature skew?

**Limitations:**

See weaknesses and questions

---

> ### Author Rebuttal · Authors · 2023-08-09
>
> >**Q1**: The model may be complex to implement, especially in environments where the number of clients is very large, due to the dual model structure (online and offline) and the knowledge transfer mechanisms.
>
> **R1**: We partially disagree with this opinion. Our method will be easy to implement if we remove the inter-client knowledge transfer mechanism. As mentioned in response to Reviewer xxVq Q7, Fed-CO$_2$ is able to address numerous non-I.I.D. challenges without the need for the inter-client knowledge transfer mechanism, and in such cases, no additional communication overhead arises as the offline model remains consistently local.
>
> >**Q2**: Despite being designed to handle heterogeneity, the performance of the framework is still reliant on the quality of the data used for training, which may not always be guaranteed, especially in a federated setting.
>
> **R2**: We respectively disagree with this comment. We believe every Deep Learning method (not limited to federated learning) relies on the quality of the training data. While our method does not rely much on the training data quality as we study very challenging data heterogeneity cases.
>
> >**Q3**: The paper does not clearly articulate how the proposed method scales with increased data, feature dimensions, or numbers of clients. Understanding the scalability of the method is crucial in real-world applications
>
> **R3**: We appreciate your advice for supplementing experiments discussing the influence of increased data, feature dimensions, and number of clients. Considering the influence of feature dimensions is rarely discussed in former FL algorithms, we have only supplemented experiments concerning increased data and client number.
>
> As for the experiments of increased data, please refer to the response to Reviewer xxVq Q5. Here, we provide the experiment results of our method with the increased client number. To be specific, we employ dataset CIFAR-10 with Dirichlet setting ($\alpha=0.3$) to conduct a series of experiments on benchmark methods with client number N $\in $ \{50, 100, 150, 300, 500\}. As Table 1 (in our new supplementary) demonstrates, our Fed-CO$_2$ consistently outperforms a bunch of SOTA methods in every case. The results in this Table and Fig. 1 (in our new supplementary) prove that our Fed-CO$_2$ owns higher scalability than benchmark methods. These supplemented experimental results will be included in our final version.
>
>
> >**Q4**: What are the computational costs associated with the proposed intra-client and inter-client knowledge transfer mechanisms?
>
> **R4**: Intra-client knowledge transfer mechanism will add one epoch in the local training phase of each client. In this epoch, the online and offline models learn from each other via knowledge distillation. Inter-client knowledge transfer mechanism does not introduce extra data epochs. The extra computation cost is the calculation of cross-entropy loss between the prediction of frozen classifiers from other clients and the labels.
>
> >**Q5**: How does Fed-CO2 handle potential security and privacy concerns associated with inter-client knowledge transfer?
>
> **R5**: Our inter-client knowledge transfer only communicates partial model parameters without exposing client's data. The traditional FL algorithm FedAvg [2] communicates clients' models to the server as well to do model parameter aggregation on the server. Therefore, model communication will not cause serious security and privacy issues.
>
> >**Q6**: How does Fed-CO2 handle scenarios where the label distribution skew and feature skew are extreme?
>
> **R6**: We enhance model cooperation via intra-client and inter-client knowledge transfer mechanisms based on the prediction fusion framework. The intra-client knowledge transfer mechanism is applied to facilitate mutual learning between the online and offline models and the inter-client knowledge transfer mechanism is utilized to leverage additional knowledge from other clients.
>
> >**Q7**:
> Can the framework handle other types of data heterogeneity beyond label distribution skew and feature skew?
>
> **R7**: Yes, our framework can handle other types of data heterogeneity. Apart from label distribution skew and feature skew, we also studied the heterogeneity case with mixed label distribution skew and feature skew. The experiment results were shown in Table 4, Table 8, 9 in the main paper.
>
> Moreover, we have added additional experiments on the new non-I.I.D. scenarios, where each client has the same label distribution but is subject to various degrees of noise. In detail, we divide the dataset CIFAR-10 into 10 clients in a way to make them independently and identically distributed. Then we apply the increasing level of random Gaussian noise with mean $\mu_{i}=0$ and standard deviation $\sigma_{i}=\frac{\sigma_M}{N-1}*i$, where $i\in$ \{0, 1,$\cdots$, N-1\}. In this series of initial experiments, we let the client number $N=10$ and the sample rate be 0.5 with $\sigma_{M}\in $ \{0, 5, 10, 15, 20\}. The backbone of each method is the same as that in other experiments.
>
> As presented in Fig. 2 (in our new supplementary), the experimental results reveal that our Fed-CO$_2$ consistently exhibits better and more robust performance across nearly all experiments, particularly in scenarios with severe noise. These experiments will be incorporated into the Appendix of our final version and will also serve as a source of inspiration for our future work.

---

> > ### Comment · Reviewer_5E9k · 2023-08-15
> >
> > Thank you for your response. It answers my questions and concerns, so I will be keeping my positive score.

---

### Official Review · Reviewer_xxVq · 2023-07-15

**Soundness:** 3 good
**Presentation:** 2 fair
**Contribution:** 3 good
**Rating:** 5
**Confidence:** 4

**Summary:**

This paper introduces an algorithm to address both label and feature heterogeneity in federated learning. The algorithm entails that each client learns two models: one that has personalized batch normalization parameters and otherwise global parameters, and another that is fully personalized. To learn the parameters, a two-stage knowledge distillation procedure is proposed, where in the first stage knowledge is distilled between the two models on each client, then in the second stage knowledge is distilled across clients. Experiments study the performance of the proposed model against a variety of baselines on image classification in the presence of label heterogeneity, feature heterogeneity, and both label and feature heterogeneity.

**Strengths:**

1. The experimental results show convincing improvement over a variety of relevant baselines in multiple settings with different types of heterogeneity. The procedure is rigorous.

2. The related works are well-discussed.

3. The paper addresses a relevant topic.

**Weaknesses:**

1. The writing can be improved. There are numerous sentences with improper grammar/semantics, e.g. “Federated learning for Data Heterogeneity of Feature Skew”.  The model parameter notation is not consistent as both \theta= \{\phi, \eta\} and \theta = \{\phi, \xi\} are used. Label and feature heterogeneity/skew are never defined. The motivation for personalizing specifically the batch normalization parameters is not clear whatsoever, and seems to be simply because it worked well in [10].

2. Label and feature heterogeneity/skew are never defined.

3. The proposed method entails learning many parameters per client. This is likely to degrade performance in settings with few samples per client, but the effect of the number of samples on algorithm performance is not investigated.

4. The experiments are limited to image classification on relatively simple datasets.

5. The idea to encourage retaining global information locally by adding a regularizer that penalizes loss on local samples of local models with concatenated heads of all other clients is interesting, but introduces additional communication and privacy concerns (due to all clients sharing each of their heads with each other) that are not addressed.

-----------

Post-rebuttal: I have raised my overall score from 4 to 5, and Contribution score from 2 to 3, please see comment below.

**Questions:**

n/a

**Limitations:**

See Weaknesses

---

> ### Author Rebuttal · Authors · 2023-08-09
>
> >**Q1**: The writing can be improved. There are numerous sentences with improper grammar/semantics, e.g. “Federated learning for Data Heterogeneity of Feature Skew”.
>
> **R1**: We will improve improper grammar/semantics in the final version.
>
>
>
> >**Q2**: The model parameter notation is not consistent as both $\theta= $\{ $\phi, \eta$ \} and $\theta = $\{ $\phi, \xi$ \} are used.
>
> **R2**: We will rectify the confusing parameter notation in our final version. The parameter notation you are confused is actually two types of parameter division. The former parameters $\theta=$\{ $\phi, \eta$ \} represent parameters of the feature extractor and the classifier (defined in section 3.1). The latter parameters $\theta=$\{$ \psi_i, \xi$ \} represents the personalized parameters and the global parameters of the model (defined in section 3.2). In the final version, we will rectify this latter notation into $\theta=$\{ $\theta_{p,i}, \theta_g$ \} to avoid misunderstanding.
>
>
> >**Q3**: Label and feature heterogeneity/skew are never defined.
>
> **R3**: Actually, we have briefly explained label distribution skew and feature skew in the Related Work (the former in line 77 and the latter in line 112). Label distribution skew is "label distribution imbalance among clients" and feature skew is "feature shift among clients (domains)".
>
> Specifically, for FL with label distribution skew, data in each client has different label distributions. For FL with feature skew, data in each client comes from different domains. In the final version, we will add more illustrations to explain label distribution skew and feature skew in the Introduction.
>
>
> >**Q4**: The motivation for personalizing specifically the batch normalization parameters is not clear whatsoever, and seems to be simply because it worked well in [10].
>
> **R4**: We respectively disagree with this comment. [10] is a classic method in addressing feature skew issues and provides a significant insight that BN helps harmonizing local feature distributions. As we mentioned in line 143, we also need to capture feature distribution in local clients. Therefore, we adopted BN personalization in our online model. Moreover, we did an ablation study in Appendix F.3, where we evaluated three personalization strategies including personalizing BNs, personalizing classification head, and personalizing BNs and classification head. The experimental results persuasively convinced us to personalize BNs.
>
>
> >**Q5**: The effect of the number of samples on algorithm performance is not investigated.
>
> **R5**: Here, we have supplemented experiments exploring the effect of the number of samples. For convenience, we utilize the Digits dataset and set 10\% of its training data as the original training dataset. Then, we select a certain ratio $\gamma$ of the original training dataset as a new training dataset for each client, where $\gamma \in $ \{0.2, 0.4, 0.6, 1.0, 2.0\}. The results are shown in Fig. 1 (in our new supplementary) and demonstrate that our Fed-CO$_2$ have an edge over various benchmark methods in almost every case, especially when the number of training data is very limited. We will add this experiment in our final version.
>
>
> >**Q6**: The experiments are limited to image classification on relatively simple datasets.
>
> **R6**: We respectively hold a different view from this comment. Actually, we utilized five distinct datasets: CIFAR-10, CIFAR-100, Digits, Office-Caltech10, and DomainNet under several non-I.I.D. settings including FL with label distribution skew, FL with feature skew, FL with label distribution skew and feature skew. Notably, most existing methods utilized even fewer datasets under more limited non-I.I.D. setting.
>
> As revealed in our paper (in line 32), prior research either concentrated on addressing label distribution skew [2, 4, 7, 11, 12, 13] with datasets CIFAR-10 and CIFAR-100, or alternatively, attempted to mitigate feature skew [8, 10, 32] with datasets Digits, Office-Caltech10, and DomainNet. Some other algorithms researched the effect of noise on FL including pFedGP [50] and [13] with datasets CIFAR-10 and CIFAR-100.
>
> Here are more details:
> 1. Experiment results for FL with label distribution skew with datasets CIFAR-10 and CIFAR-100 are shown in Table 3 (section 4.2) and Figure 4 (Appendix F.1);
> 1. Experiment results for FL with feature skew with datasets Digits, Office-Caltech10, and DomainNet are shown in Table 1, 2 (section 4.2) and Table 7 (Appendix E);
> 3. Experiment results for FL with label distribution skew and feature skew with datasets Digits, Office-Caltech10, and DomainNet are shown in Table 4 (section 4.2), Table 8, 9 (Appendix E);
> 4. We supplemented experiment results of our Fed-CO$_2$ for FL with noise on CIFAR-10 in respondence to Reviewer 5E9k Q7.
>
> [50] Achituve, Idan, et al. "Personalized federated learning with gaussian processes." Advances in Neural Information Processing Systems 34 (2021): 8392-8406.
>
> >**Q7**: The idea ... introduces additional communication and privacy concerns (due to all clients sharing each of their heads with each other) that are not addressed.
>
> **R7**: We partially disagree with this opinion. (1) Privacy concerns. In the context of traditional FedAvg [2], communicating clients' models to the server remains necessary for model parameter aggregation, meaning that model communication will not cause severe privacy concerns.
> (2) Additional communication. Our Fed-CO$_2$ merely communicates parts (the classifier) of the offline model. Thus, the additional communication overhead is relatively modest.
>
> Moreover, the inter-client knowledge transfer, which causes these two concerns, is only employed in extreme data heterogeneity issues. As our ablation study shows in Table 5, our method still achieves SOTA performance without Inter-Client Knowledge Transfer. In most non-I.I.D. cases, our framework already works.

---

> > ### Comment · Reviewer_xxVq · 2023-08-18
> > **Response**
> >
> > Thank you to the authors for your detailed response. The clarification of the experiments and new results have convinced me to raise my score. I still have concerns about the writing, and strongly encourage the authors to discuss the motivation for batch normalization in greater detail in the main body, add more clear definitions of label and feature skew -- ideally formal definitions if space permits -- and clean up grammatical mistakes.

---

### Author Rebuttal · Authors · 2023-08-09

We thank the reviewers for their careful reading of our paper and help with improving our manuscript. We sincerely appreciate that you find our work proposes 'a novel and effective approach' (Reviewer 5E9k, Reviewer NQAC), addresses 'both label distribution skew and feature skew within a unified framework' (Reviewer 5E9k), shows "promising performance" (Reviewer y4gh and Reviewer NQAC) and "convincing improvement over a variety of relevant baselines in multiple settings with different types of heterogeneity" (Reviewer xxVq), and "provided theoretical analysis" (Reviewer NQAC).

In what follows, we try to address your concerns/questions and provide a detailed item-by-item response to your comments.  According to the reviewers ' advice, we have supplemented four extra experiments to make our work more comprehensive. Specifically, we study the effect of the training number ratio and the client number on our Fed-CO$_2$ and some benchmark methods. **The results are provided in Fig.1 and Table 1 in our newly submitted attachment**. We also provide an additional ablation study about prediction fusion on FL with feature skew in our response to Reviewer NQAC Q2. Finally, we further explore the performance of our Fed-CO$_2$ and some baseline methods on FL with a new type of data heterogeneity issue caused by noise and demonstrate **the results in Fig. 2 in our newly submitted attachment**.

---

### Decision · Program_Chairs · 2023-09-21

**Decision:**

Accept (poster)

**Comment:**

The authors study the issue of data heterogeneity in Federated Learning (FL) and propose a universal FL framework that effectively handles data heterogeneity issues arising from label distribution skew, feature skew, or their combination. Specifically, they propose a novel universal cooperation framework with two models to address this challenge for both label distribution skew and feature skew data heterogeneity, referring to the model with partially personalized parameters as the online model, and the locally trained model as the offline model. They personalize Batch Normalization layers in the online model and fuse the online and offline models’ predictions as the final prediction.

The paper addresses an important problem, and the proposed framework is of interest to the FL community. In particular, the cooperation mechanism between the online and offline models is a novel and effective way to deal with data heterogeneity issues. That said, the reviewers had some concerns and questions about the paper, but they were all satisfied by the authors' responses, and they all raised their scores post-rebuttal or maintained their positive evaluations. While I think the presentation of the paper can be improved, and in particular, the motivation for the batch normalization can be further expanded in the paper, I believe the paper would be a valuable addition to the NeurIPS conference.


AC